# Genomic heterogeneity of NAD(P)H dehydrogenase predisposes *Cryptosporidium* to clofazimine resistance

Gracyn Y. Buenconsejo[1,3], Sebastian Shaw [1,3], Rui Xiao[1], Aurélia C. Balestra [1], Keenan M. O'Dea [1], Peng Jiang[2], Bingjie Xu[2], Dongqiang Wang[2], Guan Zhu [2], Daniel P. Beiting[1] & Boris Striepen [1] ✉

The parasite *Cryptosporidium* is a leading cause of life-threatening diarrhoeal disease, and effective treatment is not available. Clofazimine, an antimicrobial used for treatment of leprosy and tuberculosis, was found to have potent anti-*Cryptosporidium* activity but it failed in a human trial. This was attributed to poor bioavailability. Here we observed differential clofazimine susceptibility among *C. parvum* parasite isolates, which we exploit to identify a single genomic locus encoding the type II NADH dehydrogenase (NDH2) in an unbiased genetic cross. Targeted genetic ablation of *ndh2* resulted in high-level clofazimine resistance and biochemical studies demonstrated NDH2-mediated electron transfer to clofazimine. Through genomic analyses, we uncovered heterogeneity at the *ndh2* locus for *C. parvum* and *C. hominis*, and widespread carriage of a conserved attenuated allele across multiple continents. This heterogeneity allows parasites genomically linked through frequent sexual recombination to adjust to changing NDH2 requirements and predisposes *Cryptosporidium* to evade clofazimine treatment.

The apicomplexan parasite *Cryptosporidium* is an important cause of intestinal disease in a variety of epidemiological settings. *Cryptosporidium* infection has long been recognized as a life-threatening opportunistic infection in HIV/AIDS patients, causing watery diarrhoea and wasting associated with poor prognosis[1]. Various conditions resulting in reduced cellular immune function including certain cancers, solid organ transplantation and the accompanying immunosuppressive therapy, and multiple primary genetic defects similarly predispose individuals to life-threatening cryptosporidiosis[2]. Despite decades of effort, effective treatment is still unavailable, and the clinical management of cryptosporidiosis remains very difficult[3]. More recently, *Cryptosporidium* was also identified as a leading global cause of severe diarrhoea and associated deaths in immunocompetent young children, particularly those experiencing malnutrition[4]. In a

vicious circle, cryptosporidiosis itself predisposes children to malnutrition and stunting[5]. In the absence of a vaccine, effective drugs are urgently needed for the treatment of this large paediatric population as well[6–8]. Motivated by this, multiple large-scale screens to identify new anti-parasitic compounds were conducted[6,9]. Acknowledging the economic challenge of drug development for a largely resource-poor target population, several of these efforts attempted to leverage previous investments through repurposing of established drugs, screening leads, or cherry-picked compound libraries[10–12].

The discovery of potent anti-*Cryptosporidium* activity of clofazimine[11] was widely welcomed, as this drug has already been in clinical use for decades and is relatively inexpensive to produce. Clofazimine is a riminophenazine derivative (Fig. 1a) developed in the 1950s for the treatment of tuberculosis that was initially sidelined in favour of more

[1]Department of Pathobiology, School of Veterinary Medicine, University of Pennsylvania, Philadelphia, PA, USA. [2]State Key Laboratory for Diagnosis and Treatment of Severe Zoonotic Infectious Diseases, Institute of Zoonosis, and College of Veterinary Medicine, Jilin University, Changchun, China. [3]These authors contributed equally: Gracyn Y. Buenconsejo, Sebastian Shaw. ✉e-mail: striepen@upenn.edu

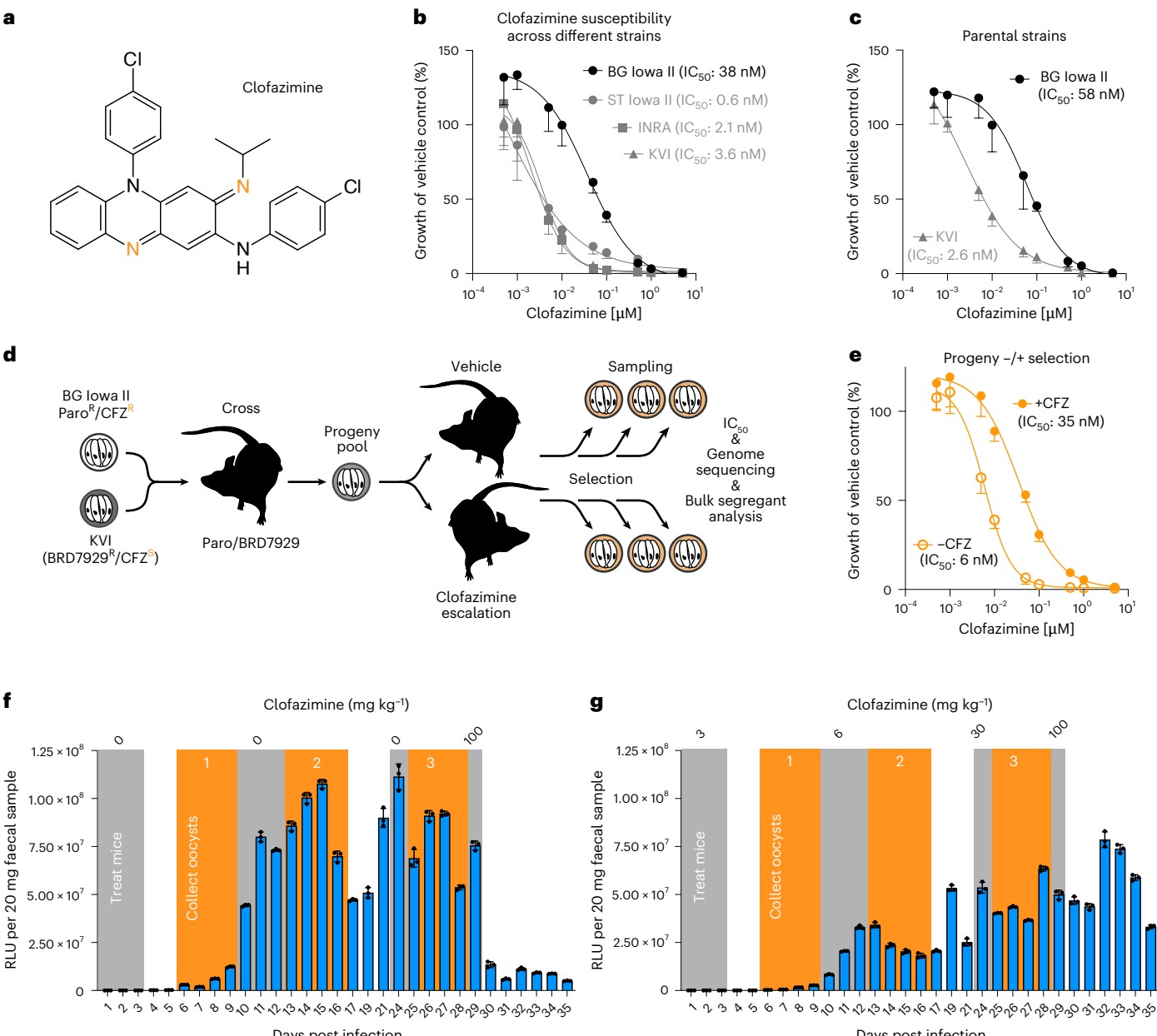

**Fig. 1 | A genetic cross of *C. parvum* strains differing in clofazimine susceptibility. a**, Chemical structure of clofazimine (CFZ). Tangerine-coloured nitrogen atoms indicate positions that accept electrons during enzymatic reduction. **b**,**c**, Parasite growth in HCT-8 tissue culture assessed by measuring luminescence in the presence of the indicated concentrations of clofazimine. Data represent mean ± s.d. of $n$ = 5 biological replicates. **d**, Experimental set up of genetic cross between BG Iowa II (CFZ-resistant) and KVI (CFZ-susceptible). Recombinant progeny was used to infect mice treated with vehicle or escalating doses of clofazimine. Oocysts were sampled throughout the selection for $IC_{50}$ determination and genome sequencing. **e**, HCT-8 culture clofazimine susceptibility assay of cross progeny treated with vehicle or drug (mean ± s.d. of $n$ = 5 biological replicates). **f**,**g**, Parasite burden was measured by following faecal luciferase activity for progeny treated with vehicle (**f**) or clofazimine (**g**); mean ± s.d., $n$ = 3 mice per cross. Grey boxes indicate treatment windows. Orange boxes indicate times oocyst were collected. No luciferase activity measurements were taken on days 18, 20, 22 and 23.

potent antibiotics[13]. However, the drug has been in widespread use for the treatment of leprosy[14]. Recently, multidrug resistance has led to a renewed use of clofazimine for treating tuberculosis[15,16]. Important questions remain in the tuberculosis field about clofazimine; most importantly, conclusive understanding of its mode of action is lacking[17]. Among the mechanisms that are debated are: binding to bacterial DNA and inhibiting replication[18], interference with bacterial redox and energy balance due to competition with menaquinone as an electron acceptor for type II NAD(P)H dehydrogenase (NDH2) which is part of the bacterial respiratory chain[19,20], or interference with bacterial potassium transport either directly or mediated through phospholipid

metabolism[21–23]. Clofazimine resistance has been reported clinically and in experimental selection in *Mycobacterium tuberculosis*, but without clear genetic clues as to the target and mode of action of the drug[17].

Most disappointingly, clofazimine failed in a human trial for the treatment of cryptosporidiosis. The trial was designed as a randomized, double-blinded, placebo-controlled study in HIV-infected adults suffering from cryptosporidiosis[24]. This was a difficult study, challenged by the very poor health of the participants, difficulties in recruitment and resulting baseline differences between study arms. Beyond the lack of efficacy, the study also observed lower than anticipated serum levels for clofazimine in the *Cryptosporidium*-infected treatment group,

which was attributed to the severity of diarrhoeal diseases in those participants. Further studies led the authors to conclude that low bioavailability of the drug[25,26] was probably responsible for the lack of efficacy.

Using a combination of forward and reverse genetic experimentation, we map clofazimine sensitivity of *Cryptosporidium* to its *ndh2* gene and population genomic studies reveal widespread genomic heterogenicity at this specific locus in both *Cryptosporidium parvum* and *Cryptosporidium hominis* genomes across the globe. These genetic findings have important consequences for the rapid emergence of resistance and provide additional clues to the interpretation of the clinical failure of clofazimine for the treatment of cryptosporidiosis. They also suggest an unconventional role of NDH2 outside of the mitochondrial respiratory chain, one that appears to benefit from the ability to maintain and modulate multiple alleles in a haploid organism through frequent sexual recombination.

## Results

### Clofazimine susceptibility varies across different *C. parvum* strains

Previous reports[11] as well as our own preliminary observations on two *C. parvum* strains led us to consider the possibility of strain-specific, heritable differences in clofazimine susceptibility. To test this more rigorously, we established the half-maximal inhibitory concentration ($IC_{50}$) for clofazimine for multiple different *C. parvum* strains. We used ST Iowa II, originally obtained from the Sterling laboratory at the University of Arizona and used in the original drug screen; BG Iowa II, a closely related strain propagated by Bunchgrass Farms and widely used for laboratory experiments; and INRA initially isolated in France[27]. All three strains are IIa genotypes derived from cattle. In contrast, KVI is a IId strain recently isolated from an infected lamb in Israel[28]. All four strains were engineered to express nanoluciferase, and we measured parasite growth over 48 h in a human ileocaecal adenocarcinoma (HCT-8) cell culture[29] and performed dose–response assays for doses ranging from 0.5 nM to 5 μM clofazimine with half-$\log_{10}$ steps (Fig. 1b, all values normalized to vehicle control for each strain). KVI, INRA and ST Iowa II showed similar susceptibility ($IC_{50}$ = 3.6 nM, 2.1 nM and 0.6 nM, with 95% confidence intervals (CI) of 2.6–4.3, 0.1–4.1, and ∞ to 3, respectively) comparable to that initially reported[11], while BG Iowa II was ~10–20 times more resistant ($IC_{50}$ = 38 nM; 95% CI = 27–52) akin to the earlier measurement for *C. hominis*[11].

### Selecting for clofazimine resistance in a genetic cross

We recently developed genetic crosses to study *Cryptosporidium*[28,30] and wondered whether we might be able to exploit this differential susceptibility to map the mechanism of action for clofazimine in this parasite in an unbiased forward genetic fashion. We chose BG Iowa II and KVI as crossing parents as they offer the largest number of distinguishing single-nucleotide polymorphisms (SNPs) and validated the susceptibility differential in strains engineered with marker cassettes suitable for a cross (neomycin phosphotransferase drug-selection marker (Neo) for BG Iowa II[29] and mutated phenylalanyl tRNA synthase[30,31] for KVI, Fig. 1c). Figure 1d outlines the design of the cross. Interferon-γ knockout (*ifnγ*−/−) mice were infected with both parental parasite strains and treated with BRD7929 and paromomycin to select for recombinant progeny carrying both resistance markers. Oocysts of this progeny were collected and used to infect two new groups of mice: one cage was treated with vehicle alone (Fig. 1f) and the other was treated with escalating doses of clofazimine (Fig. 1g), and parasite burden was measured for 35 days. This treatment regime transiently repressed parasite growth but did not cure the mice as infection rebounded each time. We collected faeces in the recovery periods (orange boxes) following each treatment (grey boxes) with drug or vehicle and isolated oocyst pools. We measured the impact of this selection scheme on clofazimine susceptibility in an in vitro dose–response assay for the final selected progeny pools

(derived from days 25–28 post infection following treatment with 30 mg kg−1 clofazimine or vehicle). The clofazimine-selected progeny showed reduction in susceptibility ($IC_{50}$ = 35 nM; 95% CI = 26–47) when compared to the vehicle-treated progeny ($IC_{50}$ = 6 nM; 95% CI = 5–7, Fig. 1e) suggesting selection for resistance.

### A single genomic locus is linked to clofazimine susceptibility

Genomic DNA was extracted from $5 \times 10^6$ oocysts of each of the three pools collected for clofazimine-selected and vehicle-treated progeny populations, and we carried out high-throughput sequencing to generate robust genome coverage (134–228-fold). Reads were aligned to the *C. parvum* genome and SNPs were called for the 4,700 positions discriminating the parental strains as detailed in Methods. Figure 2a shows allele frequencies across all 8 chromosomes compared to the *C. parvum* BG Iowa II reference genome[32], 1 indicating exclusive BG Iowa II. Clofazimine-selected populations are shown in red, vehicle in blue and later timepoints are shown in darker shades. We noted multiple peaks indicating preferred inheritance from one of the parents. These included the loci of the two selectable markers on chromosomes 3 and 5, and loci on chromosomes 2, 6 and 7 associated with enhanced virulence and persistence of the KVI parent (see our recent publication[28] for detail on these loci). However, only chromosome 7 showed differences when comparing clofazimine-treated and untreated samples (Fig. 2b, each individual SNP is represented by a dot). In the absence of treatment, KVI alleles dominated due to the virulence locus on this chromosome. Upon treatment, BG Iowa II alleles were heavily enriched pointing to preferred inheritance from the more clofazimine resistant parent. We next conducted bulk segregant analysis[28,33] to detect and measure genetic linkage. A single narrowly defined quantitative trait locus (QTL) on chromosome 7 emerged; the statistical support for this locus increased with each round of treatment and dose escalation, with the most highly significant SNP exceeding a final *G*-value of 300 (Fig. 2c,d).

### Resistance is linked to a two-base-pair deletion in the type II NAD(P)H dehydrogenase gene

The highest scoring SNP was found in gene *cgd7_1890* resulting in a valine instead of an isoleucine in a putative RNA-binding protein (Fig. 3a). This represents a conservative substitution, and while KH1 domain RNA-binding proteins can play roles in drug resistance in some cancers[34], they have not been previously associated with clofazimine resistance. We were thus intrigued to find NDH2 encoded by the next gene downstream of the SNP (*cgd7_1900*). NDH2 is one of the candidate mechanisms of clofazimine action in *Mycobacterium*[19]. However, the initial comparison of the published parental genomes showed identical *ndh2* sequences in both strains. Our bulk segregant analysis used SNPs to detect QTLs. We considered that other variations may impact drug susceptibility. This revealed a previously unrecognized INDEL—the deletion of two adenines (ΔAA) at positions 81 and 82 of the open reading frame in 100% of all reads from clofazimine-selected parasites, while only 4.5% of all reads from the vehicle-treated parasites showed a deletion at this position. Alignment of the parental genomes detected the ΔAA allele in both parents, with higher frequency in the more-resistant strain (Fig. 3b,c).

### Genetic ablation of *ndh2* confers high-level clofazimine resistance

The ΔAA deletion is predicted to result in a frameshift of the *ndh2* coding sequence and early termination of the protein after only 47 of 568 amino acids. We therefore hypothesized resistance to be the consequence of loss of NDH2 activity, rather than an SNP in *kh1*. To directly test this, we disrupted the *ndh2* gene through Cas9-directed insertion of the Nluc/Neo marker in BG Iowa II parasites using paromomycin selection (Fig. 4a). Transgenic parasites were readily obtained and PCR mapping verified appropriate insertion (Fig. 4b), and we conclude NDH2 to be dispensable. Next, we compared the sensitivity to clofazimine of the

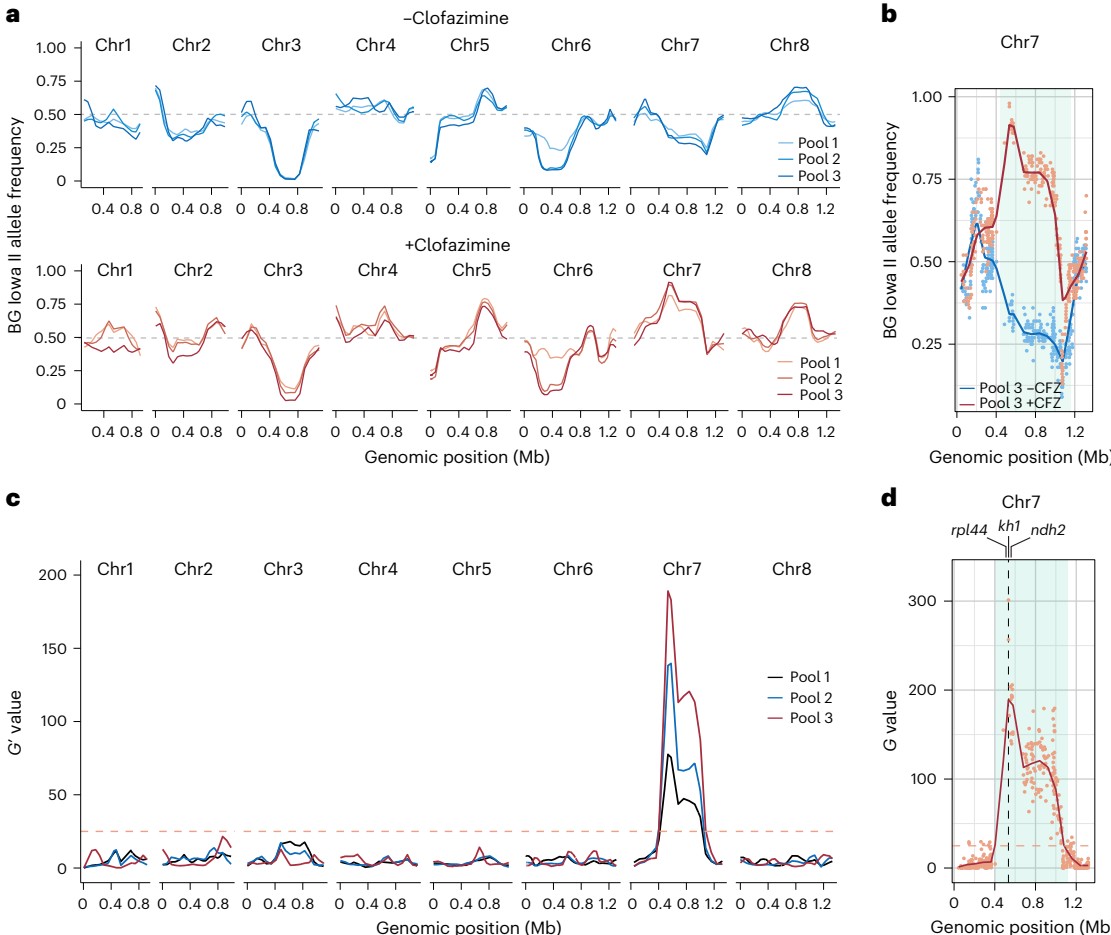

**Fig. 2 | Clofazimine susceptibility is associated with a single locus on chromosome 7. a**, Allele frequencies of SNPs distinguishing the parental strains used in the cross for all 8 *C. parvum* chromosomes. Each line represents a progeny pool; those shown in blue were treated with vehicle, those in red with clofazimine. Colour shades from light to dark indicate the 3 individual pools that were collected, sequenced and analysed. **b**, Allele frequencies of chromosome 7 for the third and final pool collected. Blue, vehicle; red, clofazimine. **c**, Whole-genome $G'$-values for genetic linkage. Lines are the weighted moving averages for the $G'$-values, and the significance threshold is shown as an orange dashed line. Dark blue, pool 1; light blue, pool 2; red, pool 3. **d**, Highly significant QTL on chromosome 7. The $G$ values of pool 3 for each individual SNP are shown as dots, the line is the weighted moving average for the $G'$-values, and the significance threshold is shown as an orange dashed line. The 95th percentile of the QTL is shown in light blue shading.

deletion mutant to wild type BG Iowa II or KVI parasites in tissue culture growth assays, and found the mutant to be highly drug resistant with an $IC_{50}$ of 4.1 µM (Fig. 4c). On the basis of these findings, we reasoned that disruption of the *ndh2* locus should be selectable by clofazimine treatment. KVI strain sporozoites were electroporated with a markerless targeting vector encoding Nluc and tdNeonGreen fluorescent protein along with a plasmid for the expression of Cas9 and a guide RNA targeting *ndh2* (Fig. 4d). Mice infected with these sporozoites were treated for 7 days with 100 mg kg$^{-1}$ day$^{-1}$ clofazimine starting at 24 h post infection. Activity of the Nluc transgene was detected in the faeces of these mice on day 4 following transfection and continued to rapidly increase (Fig. 4f). PCR analysis demonstrated transgene insertion and complete loss of the wild type locus (Fig. 4e). Oocysts subjected to flow cytometry showed uniformly bright fluorescence and when used to infect HCT-8 cultures, all parasite stages were Neon Green positive when observed by fluorescence microscopy (Fig. 4g,h). We conclude that loss of NDH2 confers resistance to clofazimine, and that clofazimine offers a new selection principle for *Cryptosporidium* transgenesis.

**Recombinant *C. parvum* NDH2 recognizes clofazimine as substrate**

Biochemical studies in bacteria have suggested that clofazimine may compete with natural quinones for electron transfer by NDH2 (the

two critical nitrogen positions are highlighted in Fig. 1a), with subsequent spontaneous reactions giving rise to reactive oxygen species that ultimately damage the cell[19]. To test this for *Cryptosporidium*, we expressed *C. parvum* NDH2 in *Escherichia coli* and purified the recombinant maltose-binding protein fusion in intact and cleaved forms (Extended Data Fig. 1a). A spectrophotometric assay was established to measure the ability of recombinant enzyme to transfer electrons from NADH to various quinone substrates[35–37]. Initial assessment showed that both intact and cleaved forms could catalyse the electron transfer from NADH to menadione with the same efficiency (Extended Data Fig. 1b,c). We observed activity with menadione (MD), menaquinone-4 (MK4) and ubiquinone-2 (CoQ2) as substrates, with low-micromolar $K_m$ values (or $K_{0.5}$ for CoQ2, which shows positive cooperativity, Fig. 4i and Extended Data Table 1). The enzyme showed highest activity with menadione ($K_m$ = 113.7 µM; $V_{max}$ = 2.0 U, U = nmol min$^{-1}$ mg$^{-1}$), and greater activity with short-chain menaquinone ($K_m$ = 139.3 µM; $V_{max}$ = 0.96 U on MK4) than with short-chain ubiquinone ($K_m$ = 7.17 µM; $V_{max}$ = 0.27 U on CoQ2). These $K_m$ (or $K_{0.5}$) values are comparable to those reported for NDH2 from other organisms, including *Saccharomyces cerevisiae* (15.2 and 7.9 µM on CoQ1 and CoQ2, respectively)[38] and *Caldalkalibacillus thermarum* (34.0 µM on MD)[39]. *C. parvum* NDH2 also reduced clofazimine, showing activity comparable to that observed for quinones ($K_m$ = 30.0 µM; $V_{max}$ = 1.0 U). Moreover, clofazimine competed with

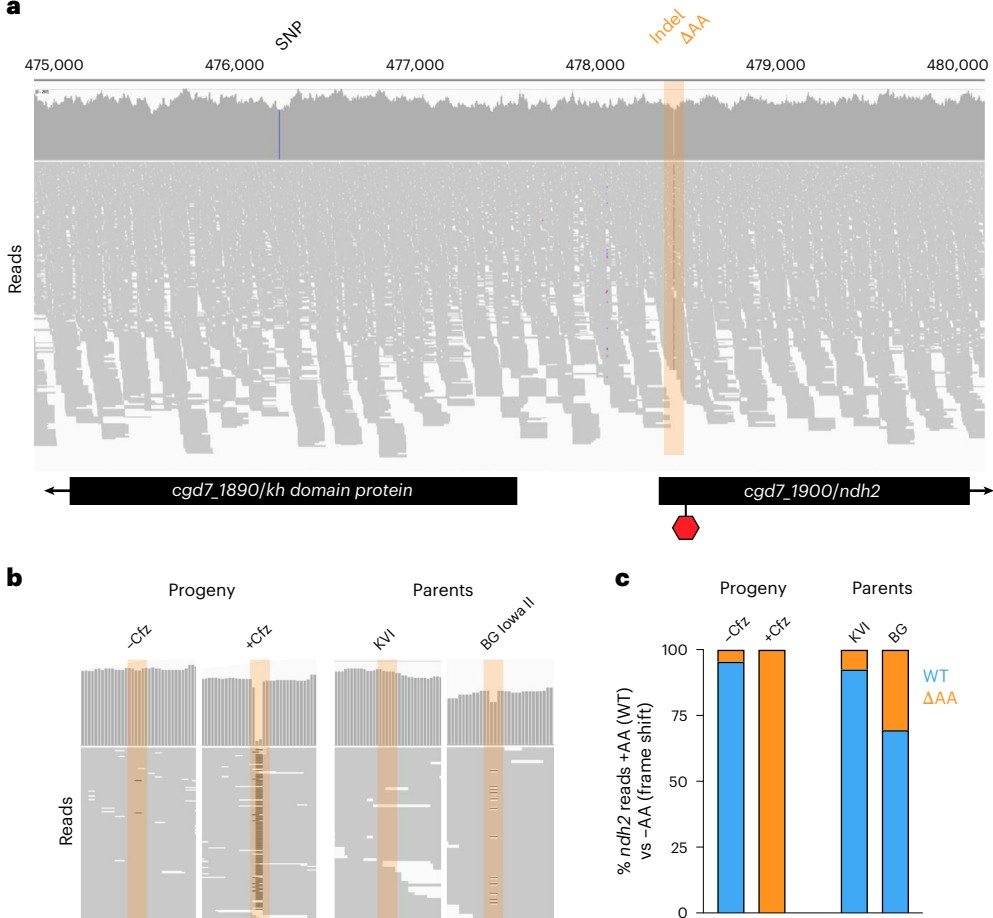

**Fig. 3 | Clofazimine-selected progeny carries an indel at the 5′ end of the *ndh2* gene. a**, Illumina sequence read alignment of the clofazimine-selected third progeny pool. Blue bar in gene *cgd7_1890* indicates the SNP with the highest *G*-value identified by bulk segregant analysis. Region highlighted in orange indicates the indel in the *ndh2* gene that introduces a frameshift and a premature stop codon at the position indicated by the red hexagon. **b**, Indel frequency comparison across the progeny either selected or not selected with clofazimine and the parental strains. Zoom-in to position 478,417–478,450 of read alignment shown. **c**, Quantification of the indel frequency represented in **b**.

native substrates, with a relative IC$_{50}$ of 2.95 µM for menadione reduction (Fig. 4j). Collectively, these biochemical experiments confirmed that *C. parvum* NDH2 indeed has type II NDH activity, and that clofazimine can be reduced by this enzyme.

## NDH2 localizes to the inner membrane complex and not the mitosome

NDH2 typically acts in the respiratory chain and in bacteria is associated with the cell membrane. In eukaryotes including related apicomplexan parasites, it is a mitochondrial protein. The *C. parvum* mitochondrion has lost its genome and much of its respiratory metabolism[40], and a small typically round organelle known as the mitosome is found in close proximity to the nucleus[41]. We were thus surprised to consistently observe NDH2-haemagglutinin (HA) staining as a line close to the surface of parasites, regardless of whether we expressed a transgene or tagged the native locus. High resolution expansion microscopy of infected HCT-8 cultures showed labelling in all intracellular stages of the parasite (Fig. 5a). NDH2 labelling outlined extracellular merozoites and male gametes during the intracellular assembly. In non-dividing parasites, the label appeared as a sharply delineated cap underlying the membrane facing away from the host cell (Fig. 5a, single arrowhead indicates this cap in a female gamete). Higher magnification reveals this labelling to coincide with the inward facing membrane of the inner membrane complex (IMC, Fig. 5b)[42]. We confirmed IMC assignment by co-labelling with an antibody to the conserved alveolin domain

of IMC proteins[43,44] (Fig. 5c, this epitope did not tolerate the expansion protocol). For comparison, we also introduced an epitope tag into *cgd8_380* which encodes malate oxidoreductase, a presumptive mitosomal protein[45]. For this protein, we indeed observed localization to a small organelle close to the nucleus (Fig. 5d). We find labelling in all stages except for male gametes, matching recent findings for alternative oxidase[46]. We conclude that NDH2 in *C. parvum* is not a mitochondrial protein but is localized to the membrane of the IMC facing the parasite cytoplasm (we note similar recent observations by Deng and colleagues[47]). Further studies are required to evaluate the nature and impact of this localization.

## Genomic heterogeneity at the *ndh2* locus and the *ΔAA* allele are widespread

We were initially surprised to find that both parental strains used in the cross carried the *ΔAA* allele, albeit at different frequencies (30.5% in BG Iowa II and 7.4% in KVI, Fig. 3b,c). This motivated a broader analysis in which we analysed publicly available whole genomic sequencing (WGS) data from 71 *C. parvum*, *C. hominis* and *C. meleagridis* sequencing read archives. These were selected for robust genome coverage (>30× mean depth, with mean mapping quality ≥60) and broad geographic representation (Fig. 6a,b). Reads were aligned to the appropriate reference genome to call variants (INDELs and SNPs with a rigorous quality score ≥20 and depth ≥20), and the frequency of the *ΔAA* allele was scored. Remarkably, the same *ΔAA* variant is detectable in the *ndh2*

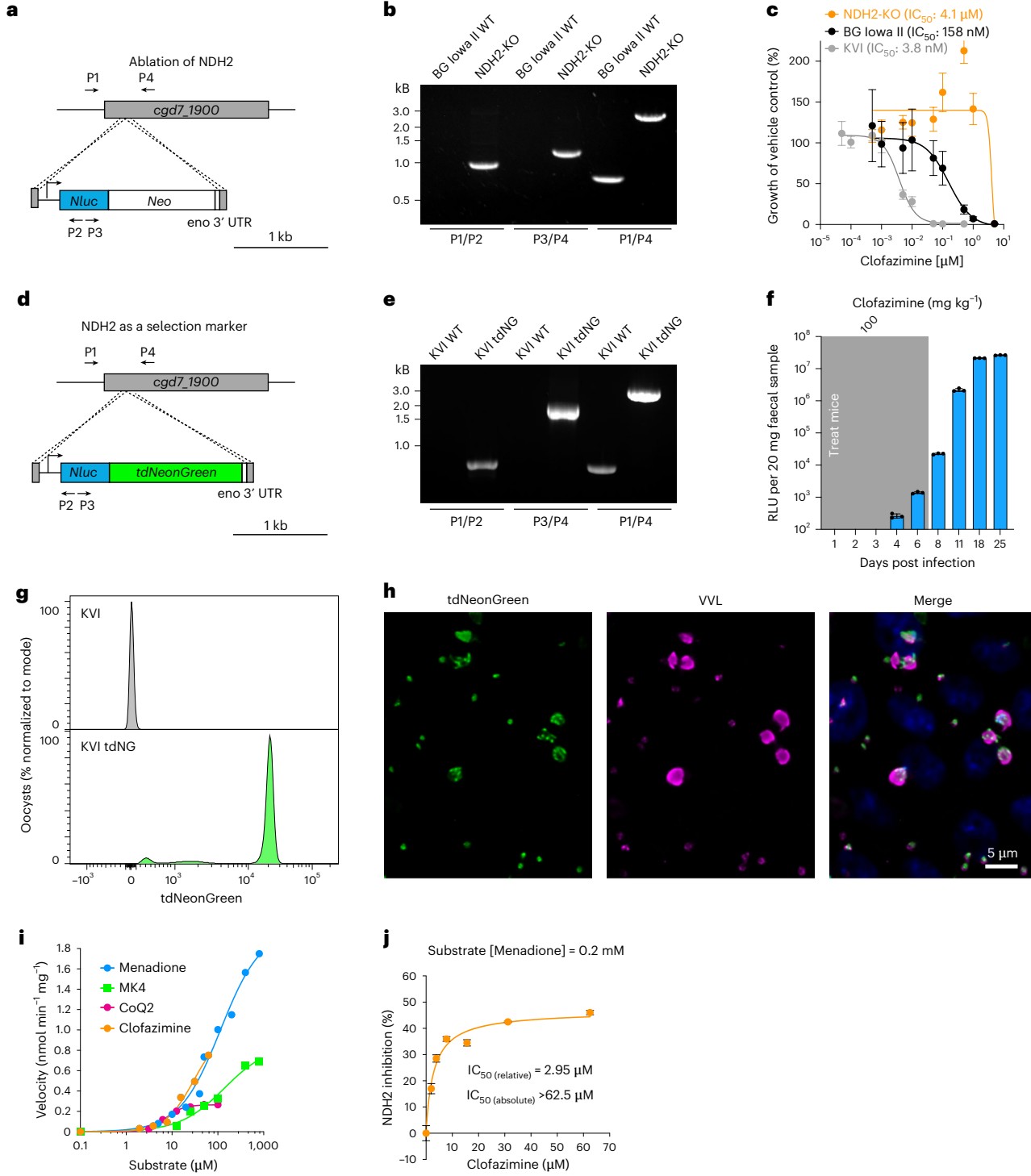

**Fig. 4 | NDH2 ablation results in high-level clofazimine resistance. a**, Genetic ablation of *ndh2* by Cas9-mediated marker insertion. P1–P4 indicate positions of primers used for PCR mapping (**b**) of the mutant and parental wildtype (WT) strains. **c**, Clofazimine IC$_{50}$ determination in HCT-8 culture of the NDH2 knockout and wildtype strains used in the cross. Data are mean ± s.d. of *n* = 5 biological replicates. **d**, Strategy and constructs used to test whether NDH2 could be used as a selection marker for *C. parvum*. P1–P4 indicate positions of primers used for PCR mapping (**e**) of the resulting transgenics compared to KVI wildtype. **f**, Luciferase activity of parasites transfected as shown in **d** and selected with clofazimine (mean ± s.d., *n* = 4 mice). No luciferase activity measurements were taken on days 5, 7, 9, 10, 12–17 and 19–24. **g**, Oocysts of clofazimine-selected transgenic mice were purified from faeces and subjected to flow cytometry (grey, KVI WT; green, KVI tdNeonGreen) or used to infect HCT-8 cells (**h**) and observed by fluorescence microscopy at 48 h of culture. Note bright green fluorescence in

flow cytometry histogram and micrograph (grey, Hoechst; green, tdNeonGreen; magenta, VVL; note that early intracellular stages stain with only modest intensity). Scale bar, 5 µm. **i**, Spectrophotometric assay of *Cp*NDH2-WT showing NADH-dependent reduction of menadione (MD), menaquinone-4 (MK4), ubiquinone-2 (CoQ2) and clofazimine. Michaelis–Menten kinetics were observed with MD and MK4, while CoQ2 and clofazimine showed positive cooperativity. Catalytic efficiency ranked MD > MK4 > CoQ2; CFZ activity was comparable to that of quinones. **j**, Clofazimine inhibits *Cp*NDH2-mediated MD reduction in a dose-dependent manner (relative IC$_{50}$ ≈ 2.95 µM; absolute IC$_{50}$ > 62.5 µM). The results support competitive reduction of clofazimine by *Cp*NDH2. The data in **i** and **j** are presented as mean ± s.e.m. The assays were performed at least 3 times independently with 3 technical replicates, and the presented data were derived from a representative experiment. Primers used for genotyping (**b**,**e**) can be found in Supplementary Data 3.

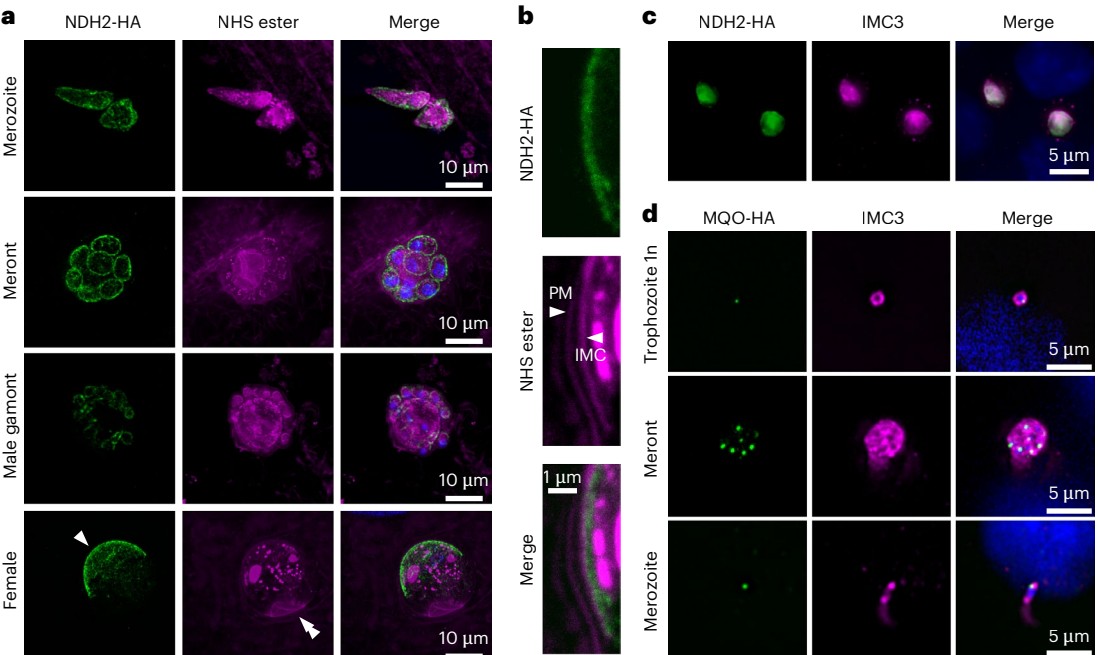

**Fig. 5 | NDH2 is localized to the inner membrane complex. a,** Expansion microscopy of HCT-8 cultures infected with NDH2-HA parasites showing protein localization throughout the life cycle. HA, green; NHS ester, magenta; Hoechst, blue; scale bar, 10 μm. Single arrowhead, NDH2 at the IMC; double arrowhead, host–parasite interface at the base of the intracellular parasites in side view. Representative micrographs of 2 independent repeats. **b,** Higher magnification detail of the female parasite shown in **a**. PM, parasite membrane; IMC, inner membrane complex; HA, green; NHS ester, magenta; Hoechst, blue; scale bar, 1 μm. Note labelling on the cytoplasmic side of the inner face of the IMC. **c,** Widefield image of NDH2-HA cells labelled with antibodies to HA and IMC3, showing co-localization. HA, green; IMC3, magenta; scale bar, 5 μm. Representative micrographs of 2 independent repeats. **d,** Widefield image of MQO-HA parasites labelled with antibodies to HA and IMC3, showing punctate MQO localization consistent with targeting to the mitosome. HA, green; IMC3, magenta; Hoechst, blue; scale bar, 5 μm. Representative micrographs of 2 independent repeats.

genes of most *C. parvum* and *C. hominis* genomes analysed (91%) (see Supplementary Data 1 for metadata). In most strains, we found a frequency of ~5–10%, while BG Iowa II stands out (note that this genome was sequenced multiple times). We wondered whether this frequency may simply reflect a broader tendency of *Cryptosporidium* genomes for hypermutation and heterogeneity. As a control, we scored all high-impact variations for *ndh2* along with 3 essential and 3 dispensable genes across 38 *C. parvum* WGS sequence read archives (SRAs) of diverse geographic origin. Figure 6c shows incidence of variation as a heat map normalized into 10 bins for each gene's coding sequence region to account for difference in length. The 5′-end of *ndh2* clearly stands out, and inspection showed the *ΔAA* allele to account for all this variation.

### The *ΔAA* allele attenuates but does not fully ablate NDH2 activity

Analysis of the *ndh2* gene shows the *ΔAA* allele to break the reading frame; however, the use of an alternative start codon might result in a largely intact enzyme with an altered N terminus (Fig. 6d). To better understand the impact of the *ΔAA* allele, we engineered two pairs of parasite strains to differ at this specific position of the gene. These were constructed either by editing the native *ndh2* locus to homogeneously wild type (WT) or ΔAA in the BG Iowa II strain, which incurred a small change in the protein sequence as well, or alternatively by complementing the knockout mutant with wildtype *ndh2* or *ndh2-ΔAA* transgenes inserted into a neutral locus (see Extended Data Fig. 2 and Methods for detail). All genotypes were validated by PCR analysis and sequencing (Fig. 6e). Engineering these strains, we also introduced HA-epitope tags allowing measurement of protein abundance. When imaging infected cultures by immunofluorescence, WT NDH2 is readily observed in all intracellular stages; the ΔAA signal is much lower but is detectable above the level of an untagged control (Fig. 6f).

Both proteins showed IMC localization. Western blot analysis of protein lysates generated from sporozoites showed a single band of the expected size for WT NDH2-HA and a much weaker, yet again detectable band for NDH2-ΔAA-HA (Fig. 6g). The predicted and measured apparent molecular weights of both proteins are near identical. Next, we conducted $IC_{50}$ determinations in culture for both strain pairs in direct comparison to NDH2-KO and found that sole expression of, or complementation with the WT allele produced clofazimine susceptibility with low nanomolar $IC_{50}$s (2.1 nM and 6 nM, 95% CI = 1.4–2.9 and 5.7–7.2, respectively). In contrast, the *ΔAA* allele conferred resistance in both backgrounds (in situ $IC_{50}$ = 494 nM; 95% CI = 292–860; ectopic $IC_{50}$ = 681 nM; 95% CI = 535–885), yet not to the full level of NDH2 deletion (Fig. 6h,i). We conclude that the *ΔAA* allele does not result in complete loss of NDH2 activity but severely attenuates its abundance leading to robust resistance.

### *ndh2* allele frequency is dynamic and responds to environmental change

We wondered how dynamic the *ndh2* locus might be and used amplicon sequencing to measure allele frequencies. We routinely purchase BG Iowa II parasites from a commercial vendor to engineer transgenic parasite. The vendor passages BG Iowa II in immunocompetent calves, while we maintain parasites in *ifnγ*$^{-/-}$ mice. We measured the *ΔAA* allele frequency across BG Iowa II samples from 2022 to 2025 and found a stable mean of 30.2 with a standard deviation of ±2.8. However, when analysing numerous transgenics derived from these parasites over this time frame, we measured a very broad distribution (Fig. 6j). This suggests that the ΔAA frequency can change, and that the specific ratio might depend on the host environment.

To test this further we explored the impact of *ndh2* heterogeneity on clofazimine susceptibility. HCT-8 cultures were infected with BG Iowa II or KVI and grown in the presence or absence of 50 nM clofazimine.

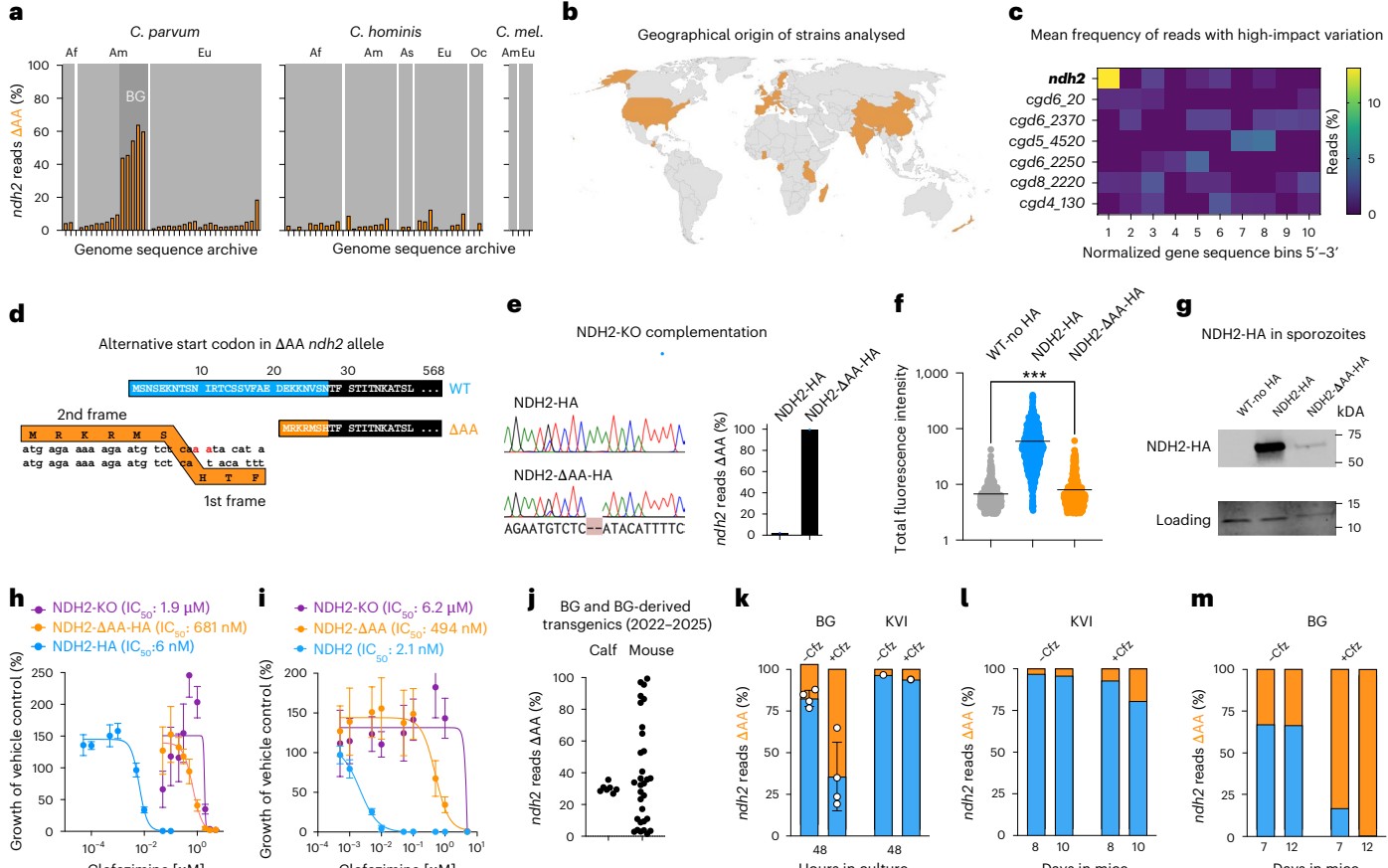

**Fig. 6 | *ndh2* heterogeneity is widespread and responds dynamically to drug pressure. a**, Analysis of *ΔAA* allele frequency across multiple publicly available whole-genome sequences of *C. parvum, C. hominis* and *C. meleagridis* strains. Each bar represents the frequency of one isolate. Af, Africa; Am, America; Eu, Europe; As, Asia; Oc, Oceania; BG, Bunchgrass Iowa II. Filters were rigorously set to a minimum of 20 unique reads mapped at this location with minimum mapping quality of 60. See Methods for detail, and all data for specific genomes are shown in Supplementary Data 1 along with accession numbers. **b**, Map showing the global origins of the samples analysed in **a** in orange. **c**, Heat map representing high-impact variations within the open reading frame of *ndh2* and 3 dispensable genes (*cgd6_20, cgd6_2370, cgd5_4520*) and 3 essential genes (*cgd6_2250, cgd8_2220, cgd4_130*). **d**, Reading frame and amino sequence at the start of NDH2 wildtype (blue) and potential alternative for the *ΔAA* allele (orange). **e**, Sanger sequencing and amplicon sequencing of NDH2-HA and NDH2 ΔAA-HA transgenics derived by complementation of NDH2-KO. **f**, Total fluorescence (mean fluorescence × area) of WT BG Iowa II parasites (*n* = 439)

lacking the HA epitope and NDH2-KO parasites expressing NDH2-HA (*n* = 638) and NDH2 ΔAA-HA (*n* = 578) in HCT-8 cultures labelled with antibody to HA. Statistical analysis was performed using one-way analysis of variance (ANOVA) and unpaired *t*-test; ***P = 0.0004. **g**, Western blot of WT sporozoites lacking the HA epitope and NHD2-KO sporozoites expressing NDH2-HA or NDH2 ΔAA-HA. Histone 3 served as loading control. Apparent molecular weights of NDH2-HA and NDH2 ΔAA-HA are 72 and 70 kDa, respectively. **h–i**, IC$_{50}$ determination comparing the complemented strains and NDH2-KO (note variance) (**h**) and the in situ HA-tagged strains homogenizing the *ndh2* locus to WT or *ΔAA* allele (**i**). Data are mean ± s.d. of *n* = 5 biological replicates. **j**, Amplicon sequencing measuring *ΔAA* allele frequencies across 5 different batches of wildtype bunchgrass farm (BG) oocysts and 33 transgenic oocysts sampled from 2022 to 2025. **k–m**, Effect of clofazimine treatment on *ΔAA* allele frequency of BG Iowa II and KVI strain parasites in tissue culture (BG; mean ± s.d., 4 biological replicates) (**k**) and in infected mice (**l,m**), measured by amplicon sequencing.

Treatment roughly doubled the ΔAA frequency in both strains over 48 h (Fig. 6k). We also surveyed the allele frequency in infected mice. In the absence of drug treatment, the *ndh2* allele frequency remained constant; in contrast, clofazimine treatment resulted in an increase in the *ΔAA* allele in both strains, reaching 100% for the BG Iowa II strain on day 10 (Fig. 6l,m). We conclude that *ndh2* heterogeneity predisposes *Cryptosporidium* to selectable drug resistance, and that the ease and speed with which resistance is achieved varies among strains and may hinge on the initial level of the *ΔAA* allele in a population.

## Discussion

Lack of effective treatment remains the crucial challenge to the clinical management of cryptosporidiosis[3,6]. The past decade has seen tremendous progress due to multiple technological advances. Leveraging the resources of the global malaria drug development effort was highly productive, and many of the strongest leads emerged from libraries

enriched for compounds that had shown promise against *Plasmodium falciparum*[12,31,48–50]. While numerous antimalarials were developed over the past century, malaria parasites have also demonstrated a remarkable ability to evade them[51]. A recent report of *Cryptosporidium* developing resistance to methionyl tRNA synthetase inhibition[52] raises concern also for this parasite.

Our study suggests that clofazimine, one of the most promising candidates for the treatment of cryptosporidiosis might have failed due to a preexisting, yet previously undetected drug-resistant allele. We identified differential susceptibility among *C. parvum* strains, matching previous observations for the single *C. hominis* strain tested[11]. Forward genetic mapping implicated the *ndh2* locus, and biochemical experiments showed the ability of NDH2 to transfer electrons to clofazimine, suggesting a prodrug activation mode of action also considered for mycobacteria[19,20]. In *Mycobacterium*, NDH2-KO did not yield drug resistance[53], but interpretation of this result is complicated by the

presence of three NDH-type enzymes. In contrast, *Cryptosporidium* has a single NDH, and we found loss of this enzyme to result in high-level resistance, and this loss is selectable by drug pressure. Importantly, a resistance-conferring allele is already present and globally distributed in the absence of drug pressure. Low bioavailability of clofazimine due to severe diarrhoea contributed to its clinical failure[24–26], and additional medicinal chemistry is likely to enhance the formulation and bioavailability of the drug[54,55]. However, our animal experimentation shows that depending on the initial allele frequency, resistance can be attained over the course of a single infection. Overall, this suggests that considering the resistance potential is an important step in future preclinical evaluation of anti-*Cryptosporidium* drugs.

NDH including type II NDH typically function as part of the respiratory chain transferring electrons to a quinone acceptor in the bacterial plasma membrane, or in eukaryotes, the inner mitochondrial membrane. In mycobacteria, the enzyme is dispensable if fatty acids are not a main carbon source[53]. In apicomplexan parasites, NDH2 replaces the canonical complex I found in many other eukaryotes[45,56]. *Toxoplasma gondii* has two NDH2 enzymes that are both dispensable[57]. The single *Plasmodium* NDH2 can be ablated with little consequence to blood stages; however, development in the mosquito is blocked[58,59]. In other apicomplexans, NDH2 is a mitochondrial enzyme, while in *Cryptosporidium* it is localized to the IMC underlying the parasite plasma membrane. This was an initially surprising finding but matches the recent spatial proteomic assignment[60]. In contrast, alternative oxidase and malate oxidoreductase, the two other redox enzymes thought to use a quinol electron carrier in *Cryptosporidium*, are indeed mitochondrial proteins (ref. 46 and Fig. 5d). The *C. parvum* mitochondrion is highly reduced, has lost its genome, the TCA cycle and most of the electron transport chain, and even those remaining elements appear dispensable[46].

The function of NDH2 at the IMC membrane is unknown, but its relocation out of the mitosome probably deemphasizes its importance to mitochondrial respiration. In the facultative intracellular pathogen *Listeria monocytogenes*, NDH2 impacts redox balance and virulence independent of the respiratory chain to adjust to different host niches[61]. NDH2 and ubiquinone might mitigate oxidative stress and balance the cytoplasmic NAD+/NADH pool. The changed localization of NDH2 could reflect a more outward-facing role in attaining and modifying critical metabolites. Multiple recent studies have found *Cryptosporidium* to interact with host- and microbiome-derived metabolites in ways that profoundly impact parasite survival[62–65]. NDH2 could play a role in detoxifying detrimental metabolites[66]. Lastly, redox pathways play key roles in antimicrobial restriction and immune signalling, and NDH2 activity may modulate these pathways[67].

Why is genomic heterogeneity at the *ndh2* locus and the specific *ΔAA* allele conserved and widespread? We propose that heterogeneity may represent a metabolic rheostat and might carry the benefit of adaptability. When BG Iowa II parasites were passaged in calves, they carried a high frequency of the attenuated *ΔAA* allele and this remained constant over years. Upon introduction into mice, this frequency varied significantly. What exactly drives this change is yet unknown, but host metabolism, host nutrition (high-fat milk replacer versus rodent chow), divergent microbiome composition, or differential immune pressure could all impact parasite redox balance. Apicomplexa are widely seen as transcriptionally hard wired with only limited metabolic flexibility[68,69]. *Cryptosporidium* is unique among apicomplexans in that it undergoes obligate sex every 2 days, resulting in very high rates of rapid recombination[70,71]. Combining this feature with genomic heterogeneity may allow the population in a single host to dial up or down a particular allele to dynamically adjust to change.

## Methods

All animal experimentation was approved by the Institutional Animal Care and Use Committee of the University of Pennsylvania (protocol 806292).

### Parasites

*C. parvum* isolates and derived transgenics used in this study were obtained and genotyped as described[28]. Original sources are: BG Iowa II, Bunchgrass Farms, Deary, Idaho; ST Iowa II, Dr Reed, University of Arizona; INRA, Dr Fabrice Laurent, INRAE and University of Tours, Nouzilly, France; KVI, Dr Yasur-Landau, Division of Parasitology, Kimron Veterinary Institution, Bet Dagan, Israel.

### Generation of transgenic strains

Guide oligo nucleotides (Sigma-Aldrich) were introduced into the *C. parvum* Cas9/U6 plasmid[29] by restriction cloning (see ref. 72 for guide design) and repair templates were constructed by Gibson assembly (New England Biolabs). Excysted sporozoites were transfected as previously described[72]. Oligos used for genotyping can be found in Supplementary Data 3.

**Ablation of *ndh2*.** The insert encodes Nluc followed by the neomycin phosphotransferase drug-selection marker and was knocked into the *ndh2* locus to induce a knockout.

Guide: 7_1900_guide_F / 7_1900_guide_R Repair template: NDH2KO_F / NDH2KO_R

***ndh2* as a selection marker.** The insert encodes Nluc followed by tdNeonGreen.

Guide: 7_1900_guide_F / 7_1900_guide_R Repair template: mNG_Cfz_F / mNG_Cfz_R

***ndh2* in situ HA-epitope.** The insert encodes a recodonized version of NDH2 (position 81 to stop codon) with a triple HA-epitope sequence followed by Nluc and the neomycin phosphotransferase drug-selection marker. 5′ homology arms between the NDH2-HA and the NDH2-ΔAA-HA strains differ. Note that the NDH2-ΔAA-HA strain has a slightly altered amino sequence at the beginning of the transgene (Fig. 6d).

NDH2-HA guide: 7_1900_guide_F / 7_1900_guide_R

NDH2-ΔAA-HA guide: NDH2_guide_2_F / NDH2_guide_2_R NDH2-HA repair template: 7_1900_repair_+AA_F / 7_1900_repair_R

NDH2-ΔAA-HA repair template: NDH2_-AA_2_F / 7_1900_repair_R

**Ectopic expression of NDH2-HA.** The inserts encode the last 113 bp of the *pheRS* gene (*cgd3_3320*, recodonized) including the mutation that confers resistance (L482V) to BRD7929. This short sequence is followed by the whole *ndh2* gene cassette (including its own promoter, and a triple HA-epitope sequence and the enolase 3′ UTR). The only difference between the repairs is the presence or absence of two adenines at position 81 (Extended Data Fig. 2b).

Guide: PheSF_guide_New_SV / PheSR_guide_New_SV Repair template: lift_NDH2_aldo3utr_F_new / lift_NDH2_3xHA_R

***cgd8_380* in situ HA-epitope.** The insert encodes a triple HA epitope followed by Nluc and the neomycin phosphotransferase drug-selection marker. We also included a reverse COWP1 3′ UTR at the 3′-end of the repair to ensure correct expression of the next downstream gene which is transcribed on the minus strand (Extended Data Fig. 2b).

Guide: 8_380_guide_F/8_380_guide_R Repair template: 8_380_tag_F_NEW/8_380_tag_R_NEW

### Cell culture and *Cryptosporidium* infections

HCT-8 cells were purchased from ATCC (CCL-224TM) and maintained in RPMI 1640 medium (Sigma-Aldrich) supplemented with 10% Cosmic calf serum (HyClone) at 37 °C in the presence of 5% $CO_2$. Oocysts were treated with 10 mM HCl at 37 °C for 45–60 min before washing and resuspension in medium containing 1% serum, 0.2 mM sodium taurocholate and 20 mM sodium bicarbonate (infection media) to induce excystation. Infection media containing oocysts were transferred immediately onto cells and remained for the duration of the infection.

## Dose–response assay and $IC_{50}$ calculations

In 96-well plates, HCT-8 cells were infected with 10,000 oocysts per well and incubated at 37 °C for 3 h. Equivalent volumes of clofazimine (2× final concentration) or 0.5% dimethylsulfoxide in infection media were added to the wells and incubated at 37 °C for 48 h. Medium was aspirated, cells were lysed and mixed with NanoGlo substrate (Promega), and luminescence was measured using a Glomax reader (Promega). $IC_{50}$ values were calculated in GraphPad Prism software v.9 (at least 2 independent experiments, each conducted with 5 replicates).

## Mouse infections and clofazimine treatment

All mouse infections were performed using 4- to 8-week-old male and female Ifnγ[−/−] (Jackson Laboratory, 002287) mice bred in-house (University of Pennsylvania). Mice were housed with a 12-h dark/light cycle, temperature between 65 and 73 °F and humidity level between 30 and 40%. Mice were pretreated with antibiotic water and infected via oral gavage as detailed in refs. [29],[72]. Clofazimine (Sigma-Aldrich, C8895) was formulated in MC-Tween (0.5% methylcellulose and 0.5% Tween-80) or PEG-glucose (70% polyethylene glycol 400 and 1.5% glucose) suspension and given to mice orally. Faeces of clofazimine- and vehicle-treated mice were collected and pooled as shown in Fig. 1f,g.

## Oocyst purification and genomic DNA extraction

Faecal materials of infected mice were collected and oocysts were purified using a sucrose gradient and CsCl flotation[72]. Genomic DNA was extracted using phenol/chloroform as previously described[30].

## Library preparation and Illumina sequencing of genomic DNA from cross progeny

We prepared Illumina libraries from extracted genomic DNA and sequenced both parents and 6 segregant pools. The library preparation was carried out using Illumina DNA Prep (former Nextera DNA Flex kit, Illumina). Subsequently, sequencing was performed on the Illumina NextSeq 2000 sequencer, utilizing the P2 300 cycle flowcell kit.

## Genotype calling

The recent Iowa II telomere to telomere de novo assembly[32] was used as reference genome to identify SNPs distinguishing the two parents, which were then used for bulk segregant analysis. Whole-genome sequencing reads for each library were individually mapped to the Iowa II de novo assembly using the BWA-MEM alignment algorithm with default parameters[73]. The resulting alignments were converted to SAM format, sorted into BAM format and deduplicated using Picard tools[8]. Variants for each sample were called using HaplotypeCaller from the GATK Suite and were subsequently aggregated across all samples using GenotypeGVCFs[74] (see https://github.com/ruicatxiao/Automated_Bulk_Segregant_Analysis for detailed parameters).

## Bulk segregant analysis

SNP loci with coverage below 30× in either of the compared pools were excluded from bulk segregant analysis. At each variable locus, we counted reads corresponding to the genotypes of each parent and calculated allele frequencies. Iowa II allele frequencies were plotted across the genome, and outliers were removed using Hampel's rule with a window size of 100 loci. Bulk segregant analysis was performed using the QTLseqr[75] R package. Extreme QTLs were defined as loci with false discovery rates (FDRs, Benjamini–Hochberg adjusted P values) below 0.01. A summary of the analysis and all metadata can be found in Supplementary Data 2.

## *cgd7_1900ΔAA* allele frequency analysis using publicly available SRAs

We developed a Python pipeline, sra2vcf (https://github.com/ruicatxiao/sra2vcf), that performs robust and comprehensive SNP/INDEL analysis on both local and online SRA datasets for long-read or short-read DNA/RNA sequencing. Briefly, for each sample, the pipeline uses the sra-toolkit to download SRA, and BWA is used to map reads to the genome. We used the *C. parvum* BG Iowa II reference genome[32] to map *C. parvum* and *C. hominis* reads, and the UGA_CmTU1867-BEI_1.0 *C. meleagridis* reference genome[76] to map the *C. meleagridis* reads. Mapped reads were sorted by genome coordinates using SAMTools, then duplicated reads were marked and removed with GATK suite. Bcftools mpileup was used to call variants using an INDEL detection optimized illumina-1.20 model. The output vcf were filtered by variant coverage and quality. Individual SRA samples' vcfs were aggregated to generate the final output table. The pipeline can accumulate and add new samples without reprocessing existing ones, and it intelligently checks for existing intermediate outputs to avoid redundant computations.

## High-impact variant analysis for essential and non-essential genes using publicly available SRAs

The sra2vcf pipeline generates individual VCF files that are first processed with SnpEff to annotate variant effects on protein coding frames for each sample, followed by SnpSift aggregation to identify HIGH-impact variants shared across samples. We developed the Python programs 'goi_af.py' and 'goi_cov.py' to analyse SnpEff-annotated VCF files along with a gene-of-interest list. The output reports allele frequencies and allele coverage for each HIGH-impact variant. We divided each gene's coding regions into 10 bins from 5′ to 3′ to generate comprehensive allele frequency tables for target genes of varying length. All analysis codes are publicly available in GitHub at https://github.com/ruicatxiao/cparvum_ndh2_clofazimine-resistance.

## Nanopore sequencing of amplified genomic material from cell cultures and faecal samples

Parasite genomic DNA was extracted from either cell culture supernatants or faecal samples as described in ref. [30]. PCR amplification of the *ndh2* locus was performed using PrimeStar Max v.2 (fwd primer: 5′-TCAAGTGGGGTCTCGGATG-3′; rev primer: 5′-CCCCACCCAGTACCTAAGATG-3′), followed by purification using the Bioneer AccuPrep Gel/PCR Purification kit before sequencing. Nanopore sequencing of purified PCR amplicons was conducted using a commercial service provided by Eurofins Genomics.

## Genomic sequencing analyses

Raw-read fastq files were mapped to the BG Iowa II de novo assembly using the BWA-MEM alignment algorithm with default parameters[73]. The resulting alignments were converted to SAM format, sorted into BAM format and deduplicated using Picard tools[8]. Alignments were visualized in IGV to calculate deletion frequencies. All codes used for this analysis are available in GitHub at https://github.com/ruicatxiao/cparvum_ndh2_clofazimine-resistance.

## Engineering transgenic strains

Guide oligonucleotides (Sigma-Aldrich) were introduced into the *C. parvum* Cas9/U6 plasmid by restriction cloning, repair templates were constructed by Gibson assembly (New England Biolabs)[72] and excysted sporozoites were transfected as previously described[72]. Briefly, $1.56 \times 10^7$ *Cp* BG Iowa II oocysts or $5 \times 10^6$ *Cp* KVI oocysts were incubated at 37 °C for 1 h in 10 mM HCl, followed by two washes with phosphate buffered saline (PBS) and an incubation at 37 °C for 1 h in 0.2 mM sodium taurocholate and 20 mM sodium bicarbonate to induce excystation[30]. Excysted sporozoites were electroporated and used to infect mice as previously described[30]. Integration was validated by PCR mapping and/or Sanger sequencing.

## Immunofluorescence assay

HCT-8 cells were seeded on coverslips in 24-well plates before infection. Infected coverslips were fixed with 4% paraformaldehyde (Sigma-Aldrich) in PBS for 20 min and then permeabilized with 0.25%

Triton X in PBS for 10 min at room temperature. Coverslips were then blocked with 1% bovine serum albumin (BSA) in PBS for 1 h before primary antibody (1:1,000 rat anti-HA, Roche, 11867423001, Clone 3F10; 1:500 rabbit anti-IMC3 or 1:1,000 biotinylated anti-VVL, Vector Laboratories, B-1235, ZD0509) incubation, followed by secondary antibody (1:1,000 Alexa Fluor 488 anti-rat, Invitrogen, A-11006, 2048174; 1:1,000 Alexa Fluor 594 anti-rabbit, Invitrogen, A-11012, 2616076; or 1:1,000 Alexa Fluor 594 streptavidin, Invitrogen, S11227, 1872019) incubation, both for 1 h in 1% BSA. Coverslips were mounted using fluoro-gel mounting medium (Electron Microscopy Sciences) and imaged using widefield Leica DM6000B or GE DeltaVision OMX microscope systems.

### Ultrastructure expansion microscopy
Ultrastructure expansion microscopy was applied to *Cryptosporidium*-infected HCT-8 cells as previously described for sporozoites[60]. Infected coverslips were fixed in 4% paraformaldehyde for 20 min at 25 °C and washed 3 times with PBS before incubating overnight at 37 °C in acrylamide and formaldehyde to prevent protein crosslinking. Samples were embedded in a water-based gel, then denatured at 95 °C before expansion in water to 4–5× their original size. Gels were shrunk in PBS before blocking and staining to save reagents. After re-expansion in water, the gels were imaged using the Leica Stellaris FALCON confocal microscope.

### Biochemical assays
Recombinant *Cp*NDH2 was expressed as a maltose-binding protein (MBP) fusion using the pMAL-c5X vector (New England Biolabs). Clofazimine (purity ≥99%) and NADH (purity ≥98%) were purchased from Yuanye Bio-Technology. Menadione (purity ≥99.5%) was obtained from MedChemExpress. Ubiquinone-2 (purity ≥95%) and menaquinone-4 (purity ≥98%) were purchased from GlpBio.

For cloning of the *Cp*NDH2 gene (gene ID *cgd7_1900*), the full-length open reading frame encoding wildtype *Cp*NDH2 (*Cp*NDH2-WT) was amplified from *C. parvum* genomic DNA (*Gp60* subtype IIdA19G1) by PCR using Phanta Max Super-Fidelity DNA Polymerase (Vazyme). The primers used were *Cp*NDH2-forward (5′-cgcgatatcgtcgacggatc-cATGTCTAACTCTGAAAAGAATACTTCCAA-3′) and *Cp*NDH2-reverse (5′-agcttatttaattacctgcagTTAGTGAGAAACGTTCATTTTGTAGATT-3′). Lowercase letters indicate additional sequences included for seamless cloning. The PCR amplicon was assembled into pMAL-c5X using the LightNing DNA Assembly Mix Plus kit (BestEnzymes Biotech). Recombinant plasmids were propagated in *E. coli* TOP10, purified and sequence verified for correct insertion.

For protein expression, the verified construct was transformed into *E. coli* BL21(DE3). Cultures were grown at 37 °C to an optical density at 450 nm (OD$_{450}$) of ~0.6, then induced with 0.5 mM isopropyl-$\beta$-D-thiogalactoside at 25 °C for 6 h. Cells were collected, lysed by sonication, and recombinant proteins purified using amylose-resin columns according to manufacturer instructions (New England Biolabs). The quality and yield of purified proteins were assessed by SDS−PAGE and Bradford assay, respectively, using bovine serum albumin as the standard.

Tag-free *Cp*NDH2 was prepared by cleavage of MBP-*Cp*NDH2 with factor Xa protease (New England Biolabs). Reactions were carried out at room temperature for 24 h in 200 μl of buffer containing 20 mM Tris (pH 8.0), 100 mM NaCl, 2 mM CaCl$_2$, 200 μg MBP-*Cp*NDH2 and 4 μg factor Xa. Following cleavage, proteins were dialysed against buffer (50 mM Tris pH 8.0, 150 mM NaCl, 1 mM EDTA) at 4 °C for 12 h, and the MBP tag was removed by amylose-resin purification. MBP alone was similarly expressed and purified for use as a negative control and for background subtraction in assays.

The catalytic activity of *Cp*NDH2-WT was measured using a spectrophotometric assay adapted from published protocols[35–37]. Reactions were carried out at 25 °C in 100 μl of buffer containing 50 mM Tris (pH 7.0), 1 mM EDTA, 0.1% Triton X-100, 400 μM NADH, 4 μM *Cp*NDH2

(MBP-fusion or tagless) and substrate at the indicated concentrations. Reactions were initiated by substrate addition, and NADH oxidation was monitored at OD$_{340}$ in a microplate reader (BioTek Instruments) at 0.5-min intervals for up to 40 min.

Since MBP-tagged and tag-free *Cp*NDH2 displayed comparable activity towards MD (Extended Data Fig. 1), subsequent assays with MK4, CoQ2 and CFZ were performed using intact MBP-*Cp*NDH2-WT. Optical density data were plotted against substrate concentrations, and kinetic parameters were estimated by nonlinear regression using a sigmoidal curve fit that incorporated the Hill coefficient. This yielded $K_{0.5}$ (or $K'$) and $V_{max}$ values. When the Hill slope was close to 1.0, the reaction followed Michaelis−Menten kinetics and $K_{0.5}$ was equivalent to $K_m$.

### Western blotting
Western blot was performed using rabbit anti-HA antibody (diluted 1:1,000, Cell Signaling Technology, 3724S, Clone C29F4) and mouse anti-H3pan antibody (diluted 1:1,000, Diagenode, C15200011, 003), as well as secondary anti-rabbit IRDye 800 (LICORbio, 926-32211, D00804-07) and anti-mouse IRDye 680 (LICORbio, 926-68070, D00804-13), both diluted 1:10,000. We followed the Licor best practice protocol. Blots were imaged using an Odyssey Licor device.

### Flow cytometry
Purified oocysts (1 million) were washed and resuspended in 200 μl FACS buffer (1× PBS, 0.2% bovine serum albumin, 1 mM EDTA). Oocysts were then passed through a 70-μM mesh filter. Data were collected on a FACSymphony A3 Lite Cell Analyzer (BD Biosciences) and analysed with FlowJo v.10 software (TreeStar). Oocysts were identified by size and plotted in a histogram for mNeonGreen expression intensity. Construct integration frequency was measured by positivity for an mNeonGreen reporter.

### Reporting summary
Further information on research design is available in the Nature Portfolio Reporting Summary linked to this article.

## Data availability
Whole-genome raw sequencing data and the raw amplicon sequencing data have been deposited in the NCBI's Sequencing Read Archive database under accession numbers PRJNA1336748 and PRJNA1337473, respectively. Source data are provided with this paper.

## Code availability
All codes used for this analysis are available in GitHub at https://github.com/ruicatxiao/cparvum_ndh2_clofazimine-resistance (ref. 77).

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

## Acknowledgements

This work was supported in part by grants from the National Institutes of Health to B.S. (R01AI112427, R01AI127798), B.S. and Christopher Hunter (R01AI148249), to the Penn Vet Imaging Core (S10OD021633), Swiss National Science Foundation fellowships 402 P2BEP3_191774 and P500PB_211097 to S.S., and European Molecular Biology Organization fellowship ALTF1145-2021 to A.C.B. We thank A. Sateriale for initial contributions, and colleagues of the Department of Pathobiology, School of Veterinary Medicine, University of Pennsylvania (Philadelphia, PA, USA), J. Byerly and C. Tang for support in animal experimentation, and B. Wallbank, A. Cohen, A. Daniels, C. Tang and M. Merolle for sharing transgenic parasites. We also thank the organizations/researchers who provided parasite strains: BG Iowa II, Bunchgrass Farms, Deary, Idaho; ST Iowa II, Dr Reed, University of Arizona; INRA, F. Laurent, INRAE and University of Tours, Nouzilly, France; KVI, Yasur-Landau, Division of Parasitology, Kimron Veterinary Institution, Bet Dagan, Israel.

## Author contributions

B.S., G.Y.B. and S.S. conceptualized the project. A.C.B., K.M.O., P.J., B.X., D.W., R.X., B.S., D.P.B., G.Z., G.Y.B. and S.S. designed the methodology. Investigations were conducted by: A.C.B. (expansion microscopy, Fig. 5a–b); K.M.O. (flow cytometry, Fig. 4g); P.J., B.X. and D.W. (biochemical assays, Fig. 4i,j, Extended Data Table 1 and Extended Data Fig. 1); R.X. (allele frequency calculations, Fig. 6a–c); G.Y.B. and S.S. (all other experiments). R.X., G.Y.B., S.S., A.C.B.,

K.M.O., P.J., B.X. and D.W. conducted formal analysis. G.Y.B., S.S. and B.S. performed visualization. G.Y.B., S.S. and B.S. wrote the original manuscript draft. All authors reviewed and edited the paper. B.S. supervised the project.

## Competing interests

The authors declare no competing interests.

## Additional information

**Extended data** is available for this paper at https://doi.org/10.1038/s41564-026-02331-5.

**Correspondence and requests for materials** should be addressed to Boris Striepen.

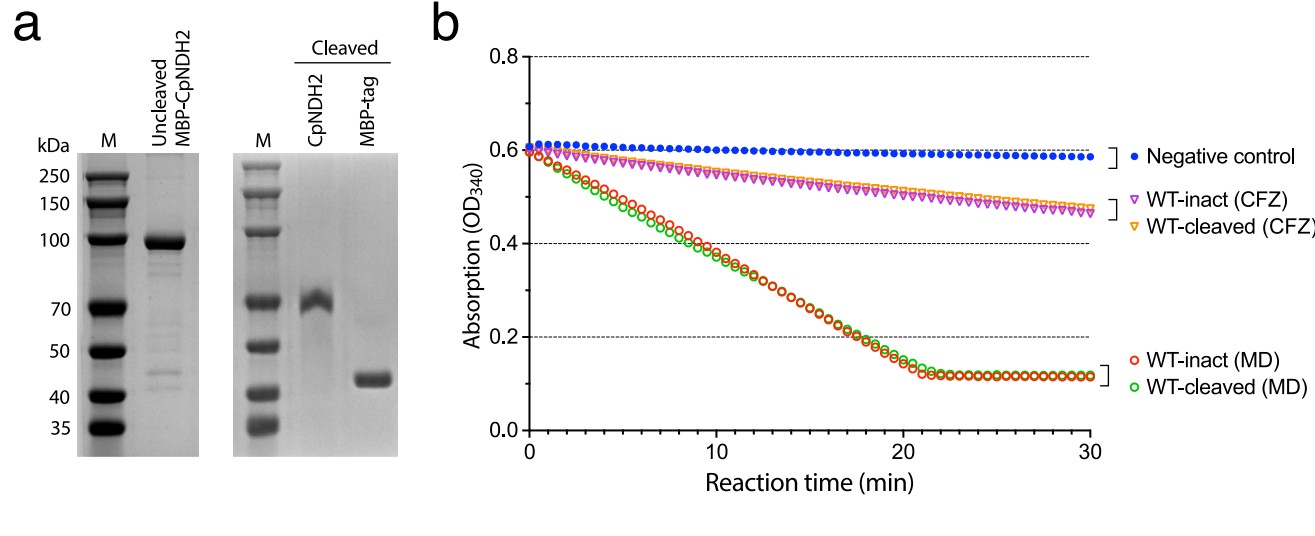

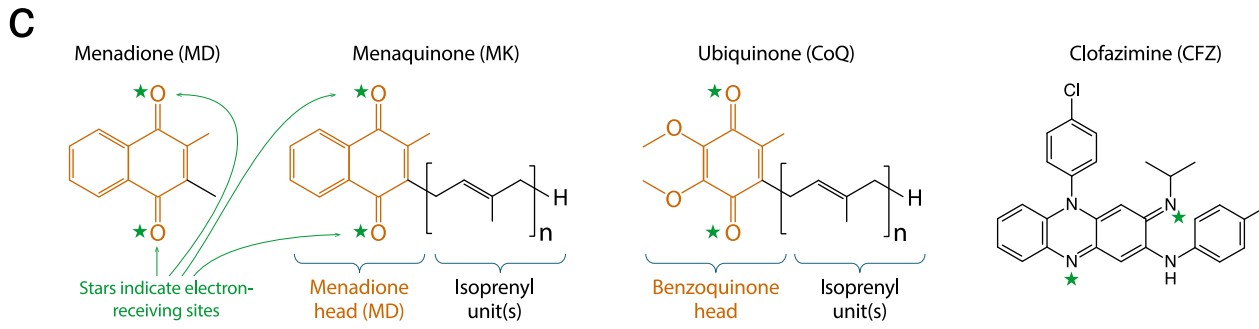

**Extended Data Fig. 1 | Initial assessment of catalytic activity using purified recombinant wild-type CpNDH2 protein in intact and cleaved forms.** (**a**) SDS-PAGE gels showing purified intact (uncleaved) and cleaved MBP-CpNDH2 protein. For each protein preparation, the quality and purity of the prepared protein were evaluated with SDS-PAGE once, stained with Coomassie Brilliant Blue (R-250). The quantity was determined by the Bradford assay using bovine serum albumin (BSA) as the standard. (**b**) Initial assessment of intact and cleaved CpNDH2 proteins in catalyzing the electron transfer from NADH to menadione (MD) and clofazimine (CFZ) using a spectrometric assay. (**c**) Chemical structures of menadione, menaquinone, ubiquinone, and clofazimine. Green stars indicate atoms positioned to accept electrons during enzymatic reduction.

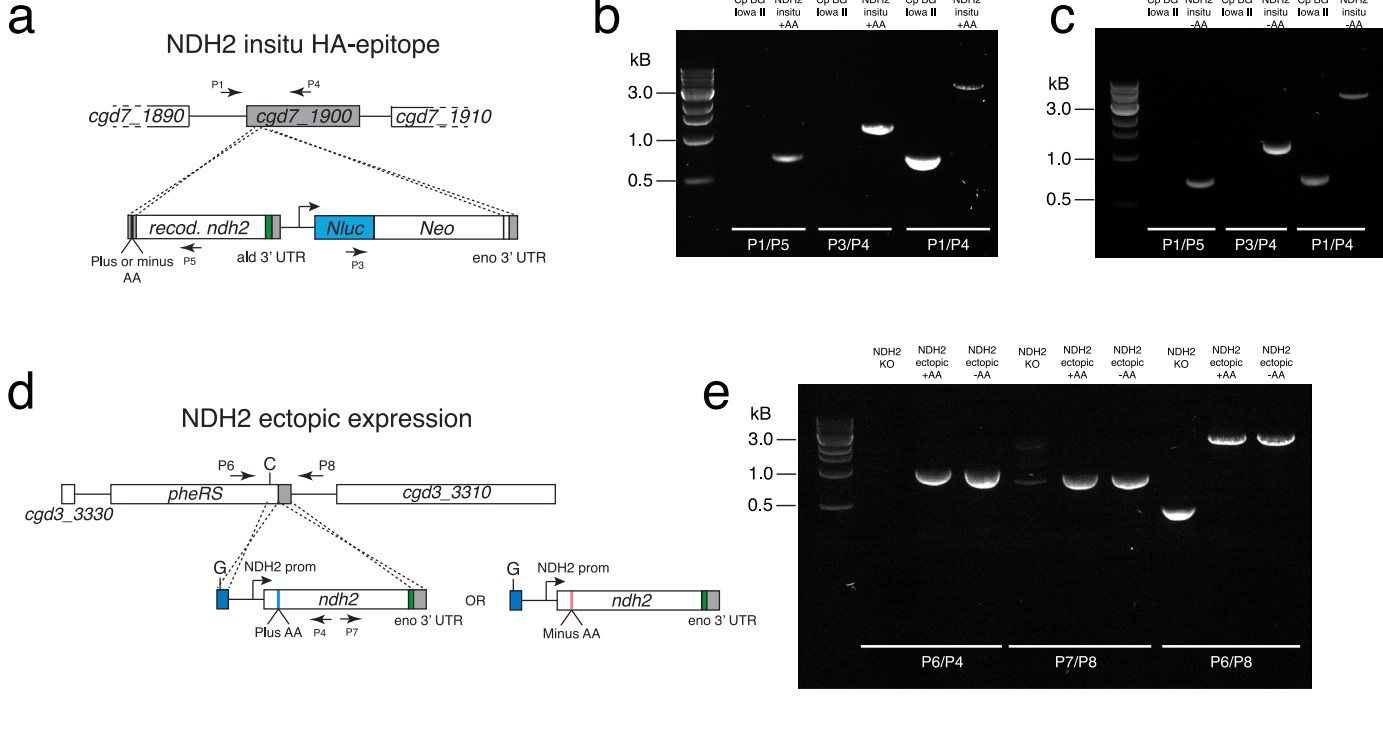

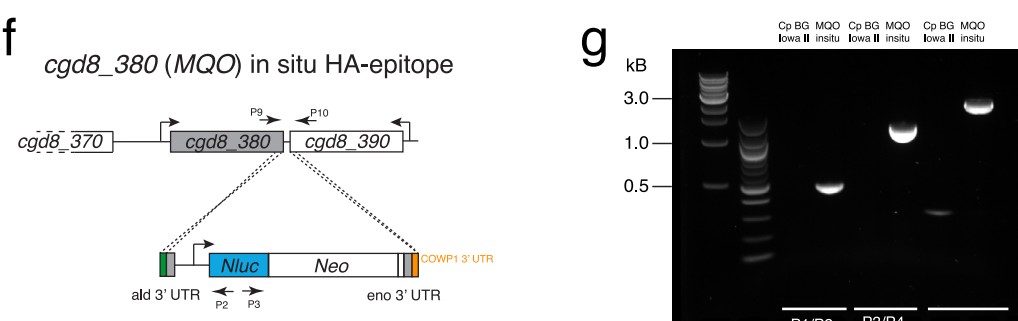

**Extended Data Fig. 2 | Transgenic parasite strains generated for this study. (a)** Map of NDH2 insitu HA epitope tagging strategy. Two strains were generated, either with or without INDEL (plus or minus AA). HA epitope = green. (**b**) Gel shows PCR mapping of the plus AA strain described in (**a**). (**c**) Gel shows PCR mapping of the minus AA strain described in (**a**). (**d**) Ectopic expression of

NDH2-HA and NDH2-ΔAA-HA. HA epitope = green. (**e**) Gel shows PCR mapping of the strains described in (**d**). (**d**). (**f**) Map of MQO insitu HA epitope tagging strategy. HA epitope = green. (**g**) Gel shows PCR mapping of the MQO insitu HA epitope strain. Labelling of the gel refers to the amplicons shown in the maps (**f**). Labelling of all the gels refer to the amplicons shown in the respective maps.

**Extended Data Table 1 | Enzyme Kinetic parameters**

| | $V_{max}$ (U)[1] | $K_{0.5}$ ($\mu$M)[2] | Hill Slope |
|---|---|---|---|
| MD | 2.00 | 113.7 | 0.97 |
| MK4 | 0.86 | 139.3 | 0.89 |
| CoQ2 | 0.27 | 7.17 | 2.24 |
| CFZ | 1.01 | 30.0 | 1.39 |

[1]U= nmol/mg/min
[2]$K_{0.5}$ = $K_m$ (when Hill slope ~1)

$V_{max}$, $K_m$ or $K_{0.5}$, Hill slope derived from nonlinear regression are summarized in the table.

# Reporting Summary

## Statistics

For all statistical analyses, confirm that the following items are present in the figure legend, table legend, main text, or Methods section.

| n/a | Confirmed | |
|---|---|---|
| ☐ | ☒ | The exact sample size (*n*) for each experimental group/condition, given as a discrete number and unit of measurement |
| ☐ | ☒ | A statement on whether measurements were taken from distinct samples or whether the same sample was measured repeatedly |
| ☐ | ☒ | The statistical test(s) used AND whether they are one- or two-sided<br>*Only common tests should be described solely by name; describe more complex techniques in the Methods section.* |
| ☒ | ☐ | A description of all covariates tested |
| ☒ | ☐ | A description of any assumptions or corrections, such as tests of normality and adjustment for multiple comparisons |
| ☐ | ☒ | A full description of the statistical parameters including central tendency (e.g. means) or other basic estimates (e.g. regression coefficient) AND variation (e.g. standard deviation) or associated estimates of uncertainty (e.g. confidence intervals) |
| ☐ | ☒ | For null hypothesis testing, the test statistic (e.g. *F*, *t*, *r*) with confidence intervals, effect sizes, degrees of freedom and *P* value noted<br>*Give P values as exact values whenever suitable.* |
| ☒ | ☐ | For Bayesian analysis, information on the choice of priors and Markov chain Monte Carlo settings |
| ☒ | ☐ | For hierarchical and complex designs, identification of the appropriate level for tests and full reporting of outcomes |
| ☒ | ☐ | Estimates of effect sizes (e.g. Cohen's *d*, Pearson's *r*), indicating how they were calculated |

*Our web collection on statistics for biologists contains articles on many of the points above.*

## Software and code

Policy information about availability of computer code

| Data collection | A Leica DM6000B Upright Widefield Fluorescence Microscope was used for timecourse and infected mouse enterocyte imaging with Leica Application Suite X version 3.7.4.23463 software.<br>A GE DeltaVision OMX Structured Illumination Super-Resolution Microscope was used for high resolution imaging of parasites with Acquire SR Acquisition control version 4.5.10296-1 software.<br>A Leica STELLARIS 8 FALCON confocal microscope with Leica LAS X software version 4.6.1.27508 was used for imaging samples prepared by ultrastructure expansion microscopy. Samples were processed using the Leica Lightning deconvolution and analyzed with the Fiji software. |
|---|---|
| Data analysis | Imaging analysis:<br>Fiji v2.26.0/1.54n<br><br>Computation analysis:<br>Bulk Segregant Analysis – BWA v0.7.17, SAMTools v1.19.2, GATk v4.6.2, magrittr v2.0.3, QTLseqr v0.7.0, ggplot2 v4.0.0, dplyr v1.1.4; R v4.42, RStudio-Server v2025.05.0-496, Automated_Bulk_Segregant_Analysis v0.1<br>Amplicon-seq Analysis - BWA v0.7.17, SAMTools v1.19.2, GATk v4.6.2, IGV v2.19.5<br>Global NDH2 Allele Frequency Analysis – sra2vcf v0.1, Python v3.10.11, SRA Toolkit v3.0.5, TrimGalore v0.6.10, BWA v0.7.17,minimap2 v2.26, STAR v2.7.10b, SAMTools v1.19.2, BCFTools v1.22, R v4.42, MultiQC v 1.25.2, RStudio-Server v2025.05.0-496<br>Global Essential_Nonessential Gene Analysis – SnpEff v5.2e, SnpSift v5.2e, R v4.42, RStudio-Server v2025.05.0-496<br>All code is available with MIT free license through Github repositories sra2vcf(https://github.com/ruicatxiao/sra2vcf) and cparvum_ndh2_clofazimine-resistance (https://github.com/ruicatxiao/cparvum_ndh2_clofazimine-resistance) |

For manuscripts utilizing custom algorithms or software that are central to the research but not yet described in published literature, software must be made available to editors and reviewers. We strongly encourage code deposition in a community repository (e.g. GitHub). See the Nature Portfolio guidelines for submitting code & software for further information.

## Data

Policy information about availability of data

All manuscripts must include a data availability statement. This statement should provide the following information, where applicable:
- Accession codes, unique identifiers, or web links for publicly available datasets
- A description of any restrictions on data availability
- For clinical datasets or third party data, please ensure that the statement adheres to our policy

> Whole genome raw sequencing data and the raw amplicon sequencing data have been deposited in the NCBI's Sequencing Read Archive database under Bioproject numbers PRJNA1336748 and PRJNA1337473, respectively.

## Research involving human participants, their data, or biological material

Policy information about studies with human participants or human data. See also policy information about sex, gender (identity/presentation), and sexual orientation and race, ethnicity and racism.

| | |
|---|---|
| Reporting on sex and gender | N/A |
| Reporting on race, ethnicity, or other socially relevant groupings | N/A |
| Population characteristics | N/A |
| Recruitment | N/A |
| Ethics oversight | N/A |

Note that full information on the approval of the study protocol must also be provided in the manuscript.

# Field-specific reporting

Please select the one below that is the best fit for your research. If you are not sure, read the appropriate sections before making your selection.

☒ Life sciences ☐ Behavioural & social sciences ☐ Ecological, evolutionary & environmental sciences

For a reference copy of the document with all sections, see nature.com/documents/nr-reporting-summary-flat.pdf

# Life sciences study design

All studies must disclose on these points even when the disclosure is negative.

| | |
|---|---|
| Sample size | Mouse experiments to isolate transgenic parasites were conducted with 4-5 mice (following Vinayak et al, Nature 523 : 477-480) and measurements of oocyst shedding used groups of 3 mice (Manjunatha et al, Nature 546: 376-380 and Shaw et al, PNAS 121: e2313210120). For microscopy-based experiments to measure HA expression we used one biological replicate and counted 10-11 randomly chosen fields per condition. n = 439 – 638 individual cells were used for the analysis. IC50s were conducted in at least 2 biological replicates, each with 5 technical replicates (Vinayak et al, Nature 523 : 477-480). |
| Data exclusions | No data were excluded |
| Replication | All attempts at replication were successful, with all IC50s repeated 2 times, except the IC50 of the non selected and selected cross (Fig 1e). |
| Randomization | Mice were initially chosen at random, but sex and age-matched between treatment and control group. Other experiments did not lend themselves to randomization due to small number of variations (e.g. plus or minus small molecule). |
| Blinding | Blinding was not used because it was not practical for a single central experimenter. Also in many experiments results were so clear that they would have subverted blinding |

# Reporting for specific materials, systems and methods

We require information from authors about some types of materials, experimental systems and methods used in many studies. Here, indicate whether each material, system or method listed is relevant to your study. If you are not sure if a list item applies to your research, read the appropriate section before selecting a response.

## Materials & experimental systems

| n/a | Involved in the study |
|---|---|
| ☐ | ☒ Antibodies |
| ☐ | ☒ Eukaryotic cell lines |
| ☒ | ☐ Palaeontology and archaeology |
| ☐ | ☒ Animals and other organisms |
| ☒ | ☐ Clinical data |
| ☒ | ☐ Dual use research of concern |
| ☒ | ☐ Plants |

## Methods

| n/a | Involved in the study |
|---|---|
| ☒ | ☐ ChIP-seq |
| ☐ | ☒ Flow cytometry |
| ☒ | ☐ MRI-based neuroimaging |

## Antibodies

**Antibodies used**

Primary Antibodies:
1 - Rat anti-HA High Affinity,from rat IgG1 (Roche, cat. 11867423001, Clone #3F10) (IFA), 1:1000 dilution
2 - Anti-MBP-IMC3 fusion protein expressed in BL21 E. coli, from rabbit antisera (IFA), 1:500 dilution
3 - Vicia Villosa Lectin (VVL, VVA), Biotinylated (Vector Laboratories, cat. B-1235, Lot #ZD0509) (IFA), 1:1000 dilution
4- HA-Tag Rabbit mAB (Cell Siganling Technology, REF 3724S, Clone #C29F4) (Western blot), 1:1000 dilution,
5- H3pan Antibody (Diagenode, REF C15200011, Lot #003) (Western blot), 1:1000 dilution,

Secondary antibodies:
IRDye-800CW goat anti-rabbit (REF 926-32211, LOT #D00804-07) (Western blot), 1:10000 dilution
IRDye-680RD goat ant-mouse (REF 926-68070, LOT #D00804-13) (Western blot), 1:10000 dilution
Goat anti-Rat IgG (H+L) Cross-Adsorbed Secondary Antibody, Alexa Fluor™ 488 (Invitrogen, cat. A-11006, Lot #2048174) (IFA), 1:1000 dilution
Goat anti-Rabbit IgG (H+L) Cross-Adsorbed Secondary Antibody, Alexa Fluor™ 594 (Invitrogen, cat. A-11012, Lot #2616076) (IFA), 1:1000 dilution
Streptavidin, Alexa Fluor™ 594 Conjugate (Invitrogen, cat. S11227, Lot #1872019) (IFA), 1:1000 dilution

**Validation**

1-HA has been used extensively in our lab: Tandel, J. et al. Life cycle progression and sexual development of the apicomplexan parasite Cryptosporidium parvum. Nat Microbiol 4, 2226-2236, doi:10.1038/s41564-019-0539-x (2019); Guerin, A. et al. Cryptosporidium uses multiple distinct secretory organelles to interact with and modify its host cell. Cell Host Microbe 31, 650-664 e656, doi:10.1016/j.chom.2023.03.001 (2023).
2-validated in Gubbels et al. (2004), https://doi.org/10.1016/j.molbiopara.2004.05.007)
3- VVL has been used extensively in the field to score Cryptosporidium: Sharling, L., et al. A Screening Pipeline for Antiparasitic Agents Targeting Cryptosporidium Inosine Monophosphate Dehydrogenase. PLoS Negl Trop Dis 4(8): e794, doi: 10.1371/journal.pntd.0000794 (2010).
4-HA has been used in our lab: Guerin, A. et al.
Cryptosporidium uses multiple distinct secretory organelles to interact with and modify its host cell. Cell Host Microbe 31, 650-664 e656, doi:10.1016/j.chom.2023.03.001 (2023)
5-Extensive validation data can be found here: https://www.diagenode.com/en/p/h3pan-monoclonal-antibody-classic-50-mg-100-ml

## Eukaryotic cell lines

*Policy information about* cell lines and Sex and Gender in Research

| | |
|---|---|
| Cell line source(s) | Human colorectal adenocarcinoma HCT-8 cells (ATCC: CCL-224TM) |
| Authentication | Cell lines not authenticated |
| Mycoplasma contamination | Cell lines were not tested for mycoplasma contamination |
| Commonly misidentified lines (See ICLAC register) | There are no commonly misidentified lines in this study |

## Animals and other research organisms

*Policy information about* studies involving animals; ARRIVE guidelines *recommended for reporting animal research, and* Sex and Gender in Research

| | |
|---|---|
| Laboratory animals | IFN-gamma knockout mice (stock number: 002287) were purchased from Jackson Laboratory and maintained as a breeding colony at the University of Pennsylvania. Mice used for experiments ranged in age from 4 to 8 weeks. Mice were housed with a 12-hour dark/light cycle, a temperature between 65 – 73 degrees Fahrenheit, and a humidity level between 30 to 40 percent. |
| Wild animals | No wild animals were used in this study. |
| Reporting on sex | Both male and female mice were used to generate and propagate Cryptosporidium parvum strains and did not exhibit a difference in |

| Reporting on sex | oocyst shedding. Mice were sex and age matched for each experiment. |
| Field-collected samples | This study involves one C.parvum strain (KVI) which was collected in the field (Shaw et al., 2025; Cell Rep; https://doi.org/ 10.1016/ j.celrep.2025.116315) |
| Ethics oversight | All animal experimentation was approved by the Institutional Animal Care and Use Committee of the University of Pennsylvania (protocol #806292). |

Note that full information on the approval of the study protocol must also be provided in the manuscript.

## Plants

| Seed stocks | N/A |
| Novel plant genotypes | N/A |
| Authentication | N/A |

## Flow Cytometry

### Plots

Confirm that:

☒ The axis labels state the marker and fluorochrome used (e.g. CD4-FITC).

☒ The axis scales are clearly visible. Include numbers along axes only for bottom left plot of group (a 'group' is an analysis of identical markers).

☒ All plots are contour plots with outliers or pseudocolor plots.

☒ A numerical value for number of cells or percentage (with statistics) is provided.

### Methodology

| Sample preparation | Oocysts were purified from fecal material |
| Instrument | BD FACSymphony™ A3 | 5-Laser Cell Analyzer |
| Software | BD FacsDiva version 9.0 |
| Cell population abundance | It is a purified sample. Mostly oocysts with some bacterial contamination. |
| Gating strategy | Oocysts were identified by size and plotted in a histogram for mNeonGreen expression intensity. |

☒ Tick this box to confirm that a figure exemplifying the gating strategy is provided in the Supplementary Information.

