## [Peer Review File · Nature Microbiology]

Genomic heterogeneity of NAD(P)H dehydrogenase predisposes *Cryptosporidium* to clofazimine resistance

Corresponding Author: Professor Boris Striepen

Version 0:

Reviewer comments:

Reviewer #1

(Remarks to the Author)

This is an absolutely outstanding manuscript that deserves immediate publication. The authors have made a major discovery about the mechanism of drug resistance in an important medically difficult to treat parasite called *Cryptosporidium* that is a major cause of diarrheal illness, and an opportunistic infection in HIV/AIDS patients. The work is very well and thoroughly done, and the paper is written very clearly. Despite decades of effort effective treatment for *Cryptosporidium* infection has remained elusive. Recently the anti-TB agent clofazimine entered human clinical trials for treatment of *Cryptosporidium* but failed, and the failure was attributed to poor bioavailability. However, in this manuscript the authors demonstrate that clofazimine resistance arises rapidly in *Cryptosporidium*. They use genetic crosses to identify mutations in type II NADH dehydrogenase (NDH2) that led to early truncation and loss of expression of active enzyme as the mechanism of resistance. They go on to use biochemical studies on recombinant enzyme to demonstrate that NDH2 mediates electron transfer to clofazimine suggesting prodrug activation is required for its mechanism of action. They show that unlike in other cells where NDH2 localizes to the mitochondria, in *C. parvum* it is localized to the membrane of the IMC facing the parasite cytoplasm, suggesting a unique role for this enzyme in the parasite. Finally, they demonstrate that the NDH2 locus of *C. parvum* and *C. hominis* has widespread heterogeneity, that the observed mutations are already present in the untreated populations and due to frequent sexual recombination in the parasite, they are predisposed to rapidly evolve resistance to clofazimine. I don't see anything that needs changing in what is the best manuscript and discovery I've read about in years.

(Remarks on code availability)

Reviewer #2

(Remarks to the Author)

Review of manuscript "Widespread genomic heterogeneity at the type II NAD(P)H dehydrogenase locus predisposes *Cryptosporidium* to clofazimine resistance" by Buenconsejo et al.

Key results

In this manuscript the authors test in vitro susceptibility to clofazimine in four *Cryptosporidium* isolates and use an in vivo cross between a susceptible and resistant strain of *C. parvum* to identify the locus conferring the resistance as a two base pair deletion in the gene for type II NAD(P)H dehydrogenase. The authors further confirmed the effect of this mutation in transgenic parasites in tissue culture growth assays. They characterize the NDH2 protein function biochemically, its localization to the inner membrane complex and that the deltaAA allele does not result in complete loss of NDH2 activity. They also show NDH2 allele frequency is dynamic and that speed of resistance development is driven by initial deltaAA frequency. Examining available 68 good quality *C. parvum* and *C. hominis* whole genomes the authors show that the deltaAA allele is widespread at low frequencies in most examined isolates.

Validity

Well described reference strains were used. Authors used an elegant method to produce genetic crosses in a IFN γ -/- mouse model and selection of drug resistant strains by clofazimine treated or untreated controls. Transgenic parasites were sequenced to verify appropriate insertions. Recombinant NDH2 production enabled biochemical analyses to verify that clofazimine can be reduced by the *C. parvum* enzyme.

Significance

The manuscript identifies a usually low-frequency NDH2 allele and its capacity for clofazimine resistance, and it brings new knowledge on the role of NDH2 localization and function in *Cryptosporidium*. The authors discuss how rapid selection of *Cryptosporidium* with the deltaAA resistance allele can rapidly result in therapy failure. The authors rightly conclude that considering resistance potential is an important step in preclinical evaluation of anti-*Cryptosporidium* drugs.

Data and methodology

The authors used *C. parvum* Ila and IId derived from animal strains for their experimental studies. This is presumably done due to available methodology and reference strains, but is also pertinent in this study considering that 11 out of 22 infections failing on Clofazimine therapy in the Tam 2021 study were infected by *C. parvum*. Supplementary data in Figure 6a helps to generalize findings to *C. hominis* and human infecting *C. parvum* strains.

I am only superficially familiar with the genetic bioinformatics pipeline used and refrain from commenting on it.

Suggested improvements

Line 80 – The Tam2021 study was randomized, but randomization in this small study failed and significant differences were found between the arms. It is therefore not quite correct to say “and a resulting lack of randomization”. I suggest to state instead f.ex “and significant baseline differences between study arms”.

It would be interesting if authors could discuss how the long half-life of clofazimine (>30 days) will affect its drug selection capability. Most of their arguments are relevant for clofazimine effect in immunosuppressed mice or humans with chronic infections. Could they also consider the effect of the deltaAA allele on treatment of the large population of immunocompetent children normally being symptomatic for a few days only?

With a starting frequency of the allele up to 5% it likely takes two or more generations to wipe out the sensitive parasites, during which one could expect temporary symptom relief and decreased shedding. The clofazimine treatment regime in the mouse model in the present manuscript transiently repressed parasite growth and shedding. In the human trial the opposite was seen in clofazimine treated patients who developed increased shedding and no change in diarrhea severity. How can this be reconciled? Could the qPCR based increase be due to shed, dead *Cryptosporidium* due to initial successful clofazimine treatment?

Interestingly, in the Tam 2021 study only 3 patients had *C. hominis*, while four patients had *C. meleagridis* and one had *C. viatorum*. Hopefully, post-hoc analyses of delta AA presence will be made in the actual strains obtained in that study. However, the authors show that the deltaAA is widespread, and present in most *parvum* and *hominis* genomes that they had access to, but not including *C. meleagridis*. With a recently available genome of *C. meleagridis* (Penumarthi et al. *Sci Data* 11, 1388 (2024) <https://doi.org/10.1038/s41597-024-04235-7>) it would be interesting to explore deltaAA presence also in this species.

Clarity and context

Abbreviation HA should be written out for clarity when first used.

Line 100 - Does the lower potency in *C. hominis* refer to ref 11? If no, a reference should be provided here. If yes, add ref 11 here.

Isolates are not coherently named. Line 122 - Authors refer to Buchgrass isolate. I suggest correcting to the name of the strain as given earlier here and in line 967. Related to this are the two ways the producer company is named. Is it “Bunch Grass” line 395 or “Bunchgrass” line 122. Also, they suddenly use “Sterling” to name a strain in line 113. I suggest to stick to defined names throughout the manuscript.

In line 110-116 authors describe a clofazimine susceptibility assay without mentioning clofazimine once. It is evident from figure 1b, but would be clearer if clofazimine was mentioned f.ex in line 111.

Can authors clarify what is meant with “Neo” in line 124.

Formatting of several references have not come out nicely and should be checked and corrected.

Conclusion

In conclusion, I think this manuscript contains a wealth of new, good quality data and a sound analysis. The conclusions are well supported by the data, and makes a significant stride forward in our knowledge of *Cryptosporidium* genetic variation and metabolism. The study is a fine example of what is now possible with new laboratory tools to examine this important enteric pathogen.

(Remarks on code availability)

Reviewer #3

(Remarks to the Author)

Clofazimine is a leprosy and TB drug that was evaluated for repurposing for treatment of cryptosporidiosis. It showed efficacy for cryptosporidiosis in pre-clinical work, but failed in clinical trials. The clinical trials were challenging for a variety of reasons (small number of patients, patients were immunocompromised, patients were suffering from severe disease), so it is difficult to determine exactly why this drug failed. One possibility is that the DMPK profile of the compound was insufficient for successful treatment in patients.

Here, the authors propose that the failure instead may be due to drug resistance. This work robustly identifies a mechanism of clofazimine resistance in *C. parvum*. However, there is no data from the failed clinical trial to confirm that the patients treated with clofazimine were infected with drug-resistant parasites.

First, the authors identify that different strains of *C. parvum* exhibit variation in susceptibility to clofazimine. They identify 2 strains that have a big variation in SNPs and that have different susceptibility to clofazimine. Using genetic tools they create a hybrid strain. The hybrid strain was already produced and characterised in another publication (PMID: 40971299); I am not sure this is very clear here.

They utilise this hybrid strain to select for resistance to clofazimine. They then use BSA and QTL mapping to identify the locus that drives resistance to clofazimine. They observe that strains that have resistance to clofazimine have an indel in the 5' region of the gene, CpNDH2. They use genetic approaches to validate that this mutation gives rise to clofazimine resistance. They

express recombinant NDH2 and confirm that it interacts with clofazimine. Typically, NDH2 should be located in the mitochondria, however *Cryptosporidium* do not have this organelle, and instead have a mitosome. Interestingly, NDH2 does not localise to the mitosome, but localises to the plasma membrane.

They also show that strains that carry the resistance-conferring mutation express this mutation with a lot of heterogeneity. Within the population of a resistant strain, there is both WT and mutant alleles; treatment with clofazimine selects for growth of the subpopulation that has the mutation.

The impact of this work is that the target of clofazimine for *Cryptosporidium* is identified and confirmed to be the major route to drug resistance. This is supported by innovative approaches including forward genetics, WGS, CRISPR for genetic validation, and enzymatic assays. This is very important clinically and also exciting as forward genetics has not yet been used to identify a drug target in *Cryptosporidium*. This technique could be useful to identify the target of other phenotypic hits in development for treatment, but that lack a known target.

It is really important to understand the mechanism of action of drugs, and especially when they fail. However, there is no evidence that this is the reason why the drug failed in the clinical trial. Samples from these patients were not sequenced to confirm that the *Cryptosporidium* in that clinical trial contained these resistance-conferring mutations. Therefore, I would caution that this study proves that the clinical trial failure was due to drug resistance.

Sometimes the figures have been oversimplified and leave out key information that is required to interpret the findings.

Specific comments:

1. All fecal Nluc graphs should be line graphs rather than bar charts. The x-axis should be a number line rather than just the dpi samples were collected. This distorts the sense of time and makes it difficult to interpret the figures. This should be changed as standard for all graphs where the x-axis is time. Technical replicates should be graphed with individual data points rather than error bars. This will make the data more clear to interpret, especially for experiments where mice were treated with a drug.
2. Report RLU/mg in all figures rather than just RLU. This has become the standard in the field.
3. Fig 1A make the nitrogens orange in color themselves rather than insert a star.
4. Fig.1B (and others), rather than use log microM on x-axis, use 10^{-x} as it is much easier to understand.
5. Fig.1B (and others), "% Luminescence of no drug" should be replaced with "% Growth inhibition (normalised to no drug control)"
6. Fig.1D all mice should be black. Use of gray mouse suggests that it's a different kind of mouse when it's just a sequential passage.
7. Fig.1F Remove gray bars when CFZ treatment was '0'. This is misleading. I know the purpose is to compare to Fig.1g but actually it makes it unclear that f is essentially the "no treatment" group and g is the treatment.
8. I find the use of blue and orange in figure 1 confusing as sometimes it's meant to indicate clofazimine and sometimes it's not. I'd rather see blue and gray as the main colors, and then orange used to indicate cfz specifically (for examples the CFZ treatment in Fig.1fg).
9. Fig.2a would benefit from having faint lines drawn at 0.5 similar to Fig.2b so it is easier to identify where regions are more similar to one of the parents. It would also be helpful to label the y-axis to indicate which parent is '0' and which is '1'.
10. Fig.2d G' is lacking the prime symbol.
11. Fig.3bd put parents on the left and progeny on the right of each panel.
12. Fig.3b is not very easy to understand what the purpose is because the sequencing maps are so small. Something zoomed in like Fig.3a for each condition would be a lot more clear.
13. Fig3. It would be helpful to label parents as KIV (CFZS) and BG (CFSR) for clarity throughout. I think the acronyms start to get complex and don't match. Sometimes it's BG and sometimes "BG Iowa II". I think it would be nice to streamline this to match.
14. Figure 4 is not color-blind friendly.
15. Fig.4ad, without indication of the size you are expecting, these drawings are not so useful in context of the DNA gels.
16. Fig.4f with the bar chart it is really difficult to interpret which days post infection mice received the dosing based on the graph alone.
17. Fig.4g y-axis of FACS data must be labelled. Gating strategy should be included as a supplement. This would also be nicely supported by microscopy.
18. Fig.4h I understand that all parasites are green, but why are not all parasites red? I think this would be better explained by showing microscopy of the oocysts rather than infected culture.
19. Figure 5 is not color-blind friendly.
20. It would be nice to quantify observations from Fig.5.
21. Fig.6a would benefit from having more details on the x-axis. As it stands I do not get a sense of how many genomic sequences were analysed (for example how many numbers of isolates, or how many isolates over time).
22. Fig.6g Why not use quantitative proteomics instead?
23. Fig.6lm and Fig.3c why don't these have data points plotted as in Fig.6k?

A few comments on the experimentation:

It would be helpful to perform some structural work to understand how NDH2 and clofazimine interact. Can the authors perform some docking studies? Or mutate regions of the recombinant protein and ablate drug interactions?

Fig.6h and i: the error bars are very big on some of these samples and the line is not accurate in representing the data. I do not think you can calculate an accurate IC50 from this data.

A few comments on the writing:

I am not sure that NDH2 "is confined to one side of the IMC" (line 359). This requires further evidence. I am not sure this can be determined using the methods applied here.

I find the phrase NHD2 “act as a functional diploid” (line 389) problematic. This is because Plasmodium and other parasites use copy-number variation as mechanism for drug resistance. The use of the word diploid here suggests a similar mechanism could occur in Cryptosporidium, where clearly there is just variation within the population, not a single parasite having a duplicated loci or a duplicated chromosome. I think this phrase should be removed for clarity.

(Remarks on code availability)

Decision Letter:

25th November 2025

Dear Boris,

Thank you for your patience while your manuscript "Widespread genomic heterogeneity at the type II NAD(P)H dehydrogenase locus predisposes Cryptosporidium to clofazimine resistance" was under peer-review at Nature Microbiology. It has now been seen by 3 referees, whose expertise and comments you will find at the end of this email. You will see from their comments below that while they find your work of interest, some important points are raised. We are very interested in the possibility of publishing your study in Nature Microbiology, but would like to consider your response to these concerns in the form of a revised manuscript before we make a final decision on publication.

In particular, you will see that while referees #1 and #2 are more positive, referee #3 says that data presentation in some of the figures should be improved and that the causative link to the failed trial needs to be toned down and clearly caveated. Please note that editorially, we will need these points to be addressed. However, we won't need further experimental work linking your results to the clinical trial, and we also overrule the request for additional structural studies on how NHD2 and clofazimine interact. The rest of the referees' reports are clear and the remaining issues should be straightforward to address.

If you have not done so already please begin to revise your manuscript so that it conforms to our Article format instructions at <http://www.nature.com/nmicrobiol/info/final-submission/>

The usual length limit for a Nature Microbiology Article is six display items (figures or tables) and 4,000 words. We have some flexibility, and can allow a revised manuscript at 4,500 words, but please consider this a firm upper limit. There is a trade-off of ~250 words per display item, so if you need more space, you could move a Figure or Table to Supplementary Information.

Some reduction could be achieved by focusing any introductory material and moving it to the start of your opening 'bold' paragraph, whose function is to outline the background to your work, describe in a sentence your new observations, and explain your main conclusions. The discussion should also be limited. Methods should be described in a separate section following the discussion, we do not place a word limit on Methods.

Nature Microbiology titles should give a sense of the main new findings of a manuscript, and should not contain punctuation. Please keep in mind that we strongly discourage active verbs in titles, and that they should ideally fit within 90 characters each (including spaces).

Please include a data availability statement as a separate section after Methods but before references, under the heading "Data Availability". This section should inform readers about the availability of the data used to support the conclusions of your study. This information includes accession codes to public repositories (data banks for protein, DNA or RNA sequences, microarray, proteomics data etc...), references to source data published alongside the paper, unique identifiers such as URLs to data repository entries, or data set DOIs, and any other statement about data availability. At a minimum, you should include the following statement: "The data that support the findings of this study are available from the corresponding author upon request", mentioning any restrictions on availability. If DOIs are provided, we also strongly encourage including these in the Reference list (authors, title, publisher (repository name), identifier, year). For more guidance on how to write this section please see: <http://www.nature.com/authors/policies/data/data-availability-statements-data-citations.pdf>

To improve the accessibility of your paper to readers from other research areas, please pay particular attention to the wording of the paper's opening bold paragraph, which serves both as an introduction and as a brief, non-technical summary in about 150 words. If, however, you require one or two extra sentences to explain your work clearly, please include them even if the paragraph is over-length as a result. The opening paragraph should not contain references. Because scientists from other sub-disciplines will be interested in your results and their implications, it is important to explain essential but specialised terms concisely. We suggest you show your summary paragraph to colleagues in other fields to uncover any problematic concepts.

If your paper is accepted for publication, we will edit your display items electronically so they conform to our house style and will reproduce clearly in print. If necessary, we will re-size figures to fit single or double column width. If your figures contain several parts, the parts should form a neat rectangle when assembled. Choosing the right electronic format at this stage will speed up the processing of your paper and give the best possible results in print. We would like the figures to be supplied as vector files - EPS, PDF, AI or postscript (PS) file formats (not raster or bitmap files), preferably generated with vector-graphics software (Adobe Illustrator for example). Please try to ensure that all figures are non-flattened and fully editable. All images should be at least 300 dpi resolution (when figures are scaled to approximately the size that they are to be printed at) and in RGB colour format. Please do not submit Jpeg or flattened TIFF files. Please see also 'Guidelines for Electronic Submission of Figures' at the end of this letter for further detail.

Figure legends must provide a brief description of the figure and the symbols used, within 350 words, including definitions of any error bars employed in the figures.

EXTENDED DATA FIGURES

Please include a statement before the acknowledgements naming the author to whom correspondence and requests for materials should be addressed.

Finally, we require authors to include a statement of their individual contributions to the paper -- such as experimental work, project planning, data analysis, etc. -- immediately after the acknowledgements. The statement should be short, and refer to authors by their initials. For details please see the Authorship section of our joint Editorial policies at http://www.nature.com/authors/editorial_policies/authorship.html

* include a point-by-point response to any editorial suggestions and to our referees. Please include your response to the editorial suggestions in your cover letter, and please upload your response to the referees as a separate document.

* ensure it complies with our format requirements for Letters as set out in our guide to authors at www.nature.com/nmicrobiol/info/gta/

* state in a cover note the length of the text, methods and legends; the number of references; number and estimated final size of figures and tables

* resubmit electronically if possible using the link below to access your home page:

Link Redacted

*This url links to your confidential homepage and associated information about manuscripts you may have submitted or be reviewing for us. If you wish to forward this e-mail to co-authors, please delete this link to your homepage first.

Please ensure that all correspondence is marked with your Nature Microbiology reference number in the subject line.

Nature Microbiology is committed to improving transparency in authorship. As part of our efforts in this direction, we are now requesting that all authors identified as 'corresponding author' on published papers create and link their Open Researcher and Contributor Identifier (ORCID) with their account on the Manuscript Tracking System (MTS), prior to acceptance. This applies to primary research papers only. ORCID helps the scientific community achieve unambiguous attribution of all scholarly contributions. You can create and link your ORCID from the home page of the MTS by clicking on 'Modify my Springer Nature account'. For more information please visit www.springernature.com/orcid.

We hope to receive your revised paper within three weeks. If you cannot send it within this time, please let us know.

Yours sincerely,

Reviewer Expertise:

Referee #1: AMR, parasite drug discovery, sequencing
Referee #2: Cryptosporidium, host-pathogen interaction
Referee #3: Cryptosporidium, drug discovery, microscopy

Reviewers Comments:

Reviewer #1 (Remarks to the Author):

This is an absolutely outstanding manuscript that deserves immediate publication. The authors have made a major discovery about the mechanism of drug resistance in an important medically difficult to treat parasite called *Cryptosporidium* that is a major cause of diarrheal illness, and an opportunistic infection in HIV/AIDS patients. The work is very well and thoroughly done, and the paper is written very clearly. Despite decades of effort effective treatment for *Cryptosporidium* infection has remained elusive. Recently the antiTB agent clofazimine entered human clinical trials for treatment of *Cryptosporidium* but failed, and the failure was attributed to poor bioavailability. However, in this manuscript the authors demonstrate that clofazimine resistance arises rapidly in *Cryptosporidium*. They use genetic crosses to identify mutations in type II NADH dehydrogenase (NDH2) that led to early truncation and loss of expression of active enzyme as the mechanism of resistance. They go on to use biochemical studies on recombinant enzyme to demonstrate that NDH2 mediates electron transfer to clofazimine suggesting prodrug activation is required for its mechanism of action. They show that unlike in other cells where NDH2 localizes to the mitochondria, in *C. parvum* it is localized to the membrane of the IMC facing the parasite cytoplasm, suggesting a unique role for this enzyme in the parasite. Finally, they demonstrate that the NDH2 locus of *C. parvum* and *C. hominis* has widespread heterogeneity, that the observed mutations are already present in the untreated populations and due to frequent sexual recombination in the parasite, they are predisposed to rapidly evolve resistance to clofazimine. I don't see anything that needs changing in what is the best manuscript and discovery I've read about in years.

Reviewer #2 (Remarks to the Author):

Review of manuscript "Widespread genomic heterogeneity at the type II NAD(P)H dehydrogenase locus predisposes *Cryptosporidium* to clofazimine resistance" by Buenconsejo et al.

Key results

In this manuscript the authors test in vitro susceptibility to clofazimine in four *Cryptosporidium* isolates and use an in vivo cross between a susceptible and resistant strain of *C. parvum* to identify the locus conferring the resistance as a two base pair deletion in the gene for type II NAD(P)H dehydrogenase. The authors further confirmed the effect of this mutation in transgenic parasites in tissue culture growth assays. They characterize the NDH2 protein function biochemically, its localization to the inner membrane complex and that the deltaAA allele does not result in complete loss of NDH2 activity. They also show NDH2 allele frequency is dynamic and that speed of resistance development is driven by initial deltaAA frequency. Examining available 68 good quality *C. parvum* and *C. hominis* whole genomes the authors show that the deltaAA allele is widespread at low frequencies in most examined isolates.

Validity

Well described reference strains were used. Authors used an elegant method to produce genetic crosses in a IFN γ -/- mouse model and selection of drug resistant strains by clofazimine treated or untreated controls. Transgenic parasites were sequenced to verify appropriate insertions. Recombinant NDH2 production enabled biochemical analyses to verify that clofazimine can be reduced by the *C. parvum* enzyme.

Significance

The manuscript identifies a usually low-frequency NDH2 allele and its capacity for clofazimine resistance, and it brings new knowledge on the role of NDH2 localization and function in *Cryptosporidium*. The authors discuss how rapid selection of *Cryptosporidium* with the deltaAA resistance allele can rapidly result in therapy failure. The authors rightly conclude that considering resistance potential is an important step in preclinical evaluation of anti-*Cryptosporidium* drugs.

Data and methodology

The authors used *C. parvum* Ila and IId derived from animal strains for their experimental studies. This is presumably done due to available methodology and reference strains, but is also pertinent in this study considering that 11 out of 22 infections failing on Clofazimine therapy in the Tam 2021 study were infected by *C. parvum*. Supplementary data in Figure 6a helps to generalize findings to *C. hominis* and human infecting *C. parvum* strains.

I am only superficially familiar with the genetic bioinformatics pipeline used and refrain from commenting on it.

Suggested improvements

Line 80 – The Tam2021 study was randomized, but randomization in this small study failed and significant differences were found between the arms. It is therefore not quite correct to say “and a resulting lack of randomization”. I suggest to state instead f.ex “and significant baseline differences between study arms”.

It would be interesting if authors could discuss how the long half-life of clofazimine (>30 days) will affect its drug selection capability. Most of their arguments are relevant for clofazimine effect in immunosuppressed mice or humans with chronic infections. Could they also consider the effect of the deltaAA allele on treatment of the large population of immunocompetent children normally being symptomatic for a few days only?

With a starting frequency of the allele up to 5% it likely takes two or more generations to wipe out the sensitive parasites, during which one could expect temporary symptom relief and decreased shedding. The clofazimine treatment regime in the mouse model in the present manuscript transiently repressed parasite growth and shedding. In the human trial the opposite was seen in clofazimine treated patients who developed increased shedding and no change in diarrhea severity. How can this be reconciled? Could the qPCR based increase be due to shed, dead *Cryptosporidium* due to initial successful clofazimine treatment?

Interestingly, in the Tam 2021 study only 3 patients had *C. hominis*, while four patients had *C. meleagridis* and one had *C. viatorum*. Hopefully, post-hoc analyses of delta AA presence will be made in the actual strains obtained in that study. However, the authors show that the deltaAA is widespread, and present in most *parvum* and *hominis* genomes that they had access to, but not including *C. meleagridis*. With a recently available genome of *C. meleagridis* (Penumarthi et al. *Sci Data* 11, 1388 (2024) <https://doi.org/10.1038/s41597-024-04235-7>) it would be interesting to explore deltaAA presence also in this species.

Clarity and context

Abbreviation HA should be written out for clarity when first used.

Line 100 - Does the lower potency in *C. hominis* refer to ref 11? If no, a reference should be provided here. If yes, add ref 11 here.

Isolates are not coherently named. Line 122 - Authors refer to Buchgrass isolate. I suggest correcting to the name of the strain as given earlier here and in line 967. Related to this are the two ways the producer company is named. Is it “Bunch Grass” line 395 or “Bunchgrass” line 122. Also, they suddenly use “Sterling” to name a strain in line 113. I suggest to stick to defined names throughout the manuscript.

In line 110-116 authors describe a clofazimine susceptibility assay without mentioning clofazimine once. It is evident from figure 1b, but would be clearer if clofazimine was mentioned f.ex in line 111.

Can authors clarify what is meant with “Neo” in line 124.

Formatting of several references have not come out nicely and should be checked and corrected.

Conclusion

In conclusion, I think this manuscript contains a wealth of new, good quality data and a sound analysis. The conclusions are well supported by the data, and makes a significant stride forward in our knowledge of *Cryptosporidium* genetic variation and metabolism. The study is a fine example of what is now possible with new laboratory tools to examine this important enteric pathogen.

Reviewer #3 (Remarks to the Author):

Clofazimine is a leprosy and TB drug that was evaluated for repurposing for treatment of cryptosporidiosis. It showed efficacy for cryptosporidiosis in pre-clinical work, but failed in clinical trials. The clinical trials were challenging for a variety of reasons (small number of patients, patients were immunocompromised, patients were suffering from severe disease), so it is difficult to determine exactly why this drug failed. One possibility is that the DMPK profile of the compound was insufficient for successful treatment in patients.

Here, the authors propose that the failure instead may be due to drug resistance. This work robustly identifies a mechanism of clofazimine resistance in *C. parvum*. However, there is no data from the failed clinical trial to confirm that the patients treated with clofazimine were infected with drug-resistant parasites.

First, the authors identify that different strains of *C. parvum* exhibit variation in susceptibility to clofazimine. They identify 2 strains that have a big variation in SNPs and that have different susceptibility to clofazimine. Using genetic tools they create a hybrid strain. The hybrid strain was already produced and characterised in another publication (PMID: 40971299); I am not sure this is very clear here.

They utilise this hybrid strain to select for resistance to clofazimine. They then use BSA and QTL mapping to identify the locus that drives resistance to clofazimine. They observe that strains that have resistance to clofazimine have an indel in the 5' region of the gene, CpNDH2. They use genetic approaches to validate that this mutation gives rise to clofazimine resistance. They express recombinant NDH2 and confirm that it interacts with clofazimine. Typically, NDH2 should be located in the mitochondria, however *Cryptosporidium* do not have this organelle, and instead have a mitosome. Interestingly, NDH2 does not localise to the mitosome, but localises to the plasma membrane.

They also show that strains that carry the resistance-conferring mutation express this mutation with a lot of heterogeneity. Within the population of a resistant strain, there is both WT and mutant alleles; treatment with clofazimine selects for growth of the subpopulation that has the mutation.

The impact of this work is that the target of clofazimine for *Cryptosporidium* is identified and confirmed to be the major route to

drug resistance. This is supported by innovative approaches including forward genetics, WGS, CRISPR for genetic validation, and enzymatic assays. This is very important clinically and also exciting as forward genetics has not yet been used to identify a drug target in *Cryptosporidium*. This technique could be useful to identify the target of other phenotypic hits in development for treatment, but that lack a known target.

It is really important to understand the mechanism of action of drugs, and especially when they fail. However, there is no evidence that this is the reason why the drug failed in the clinical trial. Samples from these patients were not sequenced to confirm that the *Cryptosporidium* in that clinical trial contained these resistance-conferring mutations. Therefore, I would caution that this study proves that the clinical trial failure was due to drug resistance.

Sometimes the figures have been oversimplified and leave out key information that is required to interpret the findings.

Specific comments:

1. All fecal NIuc graphs should be line graphs rather than bar charts. The x-axis should be a number line rather than just the dpi samples were collected. This distorts the sense of time and makes it difficult to interpret the figures. This should be changed as standard for all graphs where the x-axis is time. Technical replicates should be graphed with individual data points rather than error bars. This will make the data more clear to interpret, especially for experiments where mice were treated with a drug.
2. Report RLU/mg in all figures rather than just RLU. This has become the standard in the field.
3. Fig 1A make the nitrogens orange in color themselves rather than insert a star.
4. Fig.1B (and others), rather than use log microM on x-axis, use 10^{-x} as it is much easier to understand.
5. Fig.1B (and others), “% Luminescence of no drug” should be replaced with “% Growth inhibition (normalised to no drug control)”
6. Fig.1D all mice should be black. Use of gray mouse suggests that it’s a different kind of mouse when it’s just a sequential passage.
7. Fig.1F Remove gray bars when CFZ treatment was ‘0’. This is misleading. I know the purpose is to compare to Fig.1g but actually it makes it unclear that f is essentially the “no treatment” group and g is the treatment.
8. I find the use of blue and orange in figure 1 confusing as sometimes it’s meant to indicate clofazimine and sometimes it’s not. I’d rather see blue and gray as the main colors, and then orange used to indicate cfz specifically (for examples the CFZ treatment in Fig.1fg).
9. Fig.2a would benefit from having faint lines drawn at 0.5 similar to Fig.2b so it is easier to identify where regions are more similar to one of the parents. It would also be helpful to label the y-axis to indicate which parent is ‘0’ and which is ‘1’.
10. Fig.2d G’ is lacking the prime symbol.
11. Fig.3bd put parents on the left and progeny on the right of each panel.
12. Fig.3b is not very easy to understand what the purpose is because the sequencing maps are so small. Something zoomed in like Fig.3a for each condition would be a lot more clear.
13. Fig3. It would be helpful to label parents as KIV (CFZS) and BG (CFSR) for clarity throughout. I think the acronyms start to get complex and don’t match. Sometimes it’s BG and sometimes “BG Iowa II”. I think it would be nice to streamline this to match.
14. Figure 4 is not color-blind friendly.
15. Fig.4ad, without indication of the size you are expecting, these drawings are not so useful in context of the DNA gels.
16. Fig.4f with the bar chart it is really difficult to interpret which days post infection mice received the dosing based on the graph alone.
17. Fig.4g y-axis of FACS data must be labelled. Gating strategy should be included as a supplement. This would also be nicely supported by microscopy.
18. Fig.4h I understand that all parasites are green, but why are not all parasites red? I think this would be better explained by showing microscopy of the oocysts rather than infected culture.
19. Figure 5 is not color-blind friendly.
20. It would be nice to quantify observations from Fig.5.
21. Fig.6a would benefit from having more details on the x-axis. As it stands I do not get a sense of how many genomic sequences were analysed (for example how many numbers of isolates, or how many isolates over time).
22. Fig.6g Why not use quantitative proteomics instead?
23. Fig.6lm and Fig.3c why don’t these have data points plotted as in Fig.6k?

A few comments on the experimentation:

It would be helpful to perform some structural work to understand how NHD2 and clofazimine interact. Can the authors perform some docking studies? Or mutate regions of the recombinant protein and ablate drug interactions?

Fig.6h and i: the error bars are very big on some of these samples and the line is not accurate in representing the data. I do not think you can calculate an accurate IC50 from this data.

A few comments on the writing:

I am not sure that NDH2 “is confined to one side of the IMC” (line 359). This requires further evidence. I am not sure this can be determined using the methods applied here.

I find the phrase NDH2 “act as a functional diploid” (line 389) problematic. This is because *Plasmodium* and other parasites use copy-number variation as mechanism for drug resistance. The use of the word diploid here suggests a similar mechanism could occur in *Cryptosporidium*, where clearly there is just variation within the population, not a single parasite having a duplicated loci or a duplicated chromosome. I think this phrase should be removed for clarity.

Version 1:

Reviewer comments:

Reviewer #2

(Remarks to the Author)

My remarks have been well addresses in the rebuttal letter and in the corresponding manuscript text and figures.

Regards,
Kurt Hanevik

(Remarks on code availability)

Reviewer #3

(Remarks to the Author)

Thanks for the thoughtful replies and updates as requested. I think all but one have been sufficiently addressed.

I think my comment below was misunderstood, and I would like to explain again what I meant:

"1. All fecal Nluc graphs should be line graphs rather than bar charts. The x-axis should be a number line rather than just the dpi samples were collected."

By this I meant the x-axis should be a number line, where the numbers are not a category but a number line. This means you can tell the true distance between dpi. Right now sometimes the numbers are sequential, and sometimes there are gaps. This makes it difficult to interpret because it gives the appearance that all the data points have equal lengths of time between them. But they do not, and this complicates how to interpret things, especially when mice have been dosed.

If you prefer to keep the bar graph over the number line, that seems ok, but I think the x-axis should reflect time more appropriately.

(Remarks on code availability)

Reviewer #4

(Remarks to the Author)

This study provides a compelling genetic explanation for a critical clinical challenge in the treatment of *Cryptosporidium*: the suboptimal efficacy of clofazimine. By integrating genetic crosses, Bulk Segregant Analysis (BSA), and rigorous gene-editing validation, the authors demonstrate that the heterogeneity of NDH2 alleles is the primary driver of resistance. This finding represents a significant paradigm shift, moving beyond the traditional view that clinical failure is solely attributable to poor drug bioavailability, and instead revealing a sophisticated evolutionary strategy of pre-existing resistance within the parasite population.

The genomics analysis section of the manuscript was specifically reviewed. The authors successfully identified a AAA deletion (Δ AA) located in a highly variable region and, using a BSA strategy, precisely linked the resistant phenotype to the NDH2 gene. This approach integrates classical genetics with high-throughput sequencing, making it highly convincing. Furthermore, a systematic reanalysis of sequencing data from global isolates further confirmed the widespread presence of NDH2 heterogeneity in natural populations. The application of genomic data and the analytical strategies employed in this study are scientifically sound and logically rigorous. The findings are well-supported by the data presented.

(Remarks on code availability)

Decision Letter:

16th February 2026

Dear Boris,

Thank you for your patience while we recruited an additional referee to cover the genomics aspects of your Article entitled "Widespread genomic heterogeneity at the type II NAD(P)H dehydrogenase locus predisposes *Cryptosporidium* to clofazimine resistance". We have now received the referee's report and I am attaching it below (along the other two reports). The referee is satisfied with the robustness of the data and therefore we will be happy to in principle accept your study for publication. However, we will first need you to convert your two Supplementary Figures to Extended Data Figures (please see below). Once that is done and you resubmitted using the link below, we will proceed to initiate our checks and will then get back to you with an author checklist with further guidance on formatting.

EXTENDED DATA FIGURES

Link Redacted

Nature Microbiology is committed to improving transparency in authorship. As part of our efforts in this direction, we are now requesting that all authors identified as 'corresponding author' on published papers create and link their Open Researcher and Contributor Identifier (ORCID) with their account on the Manuscript Tracking System (MTS), prior to acceptance. This applies to primary research papers only. ORCID helps the scientific community achieve unambiguous attribution of all scholarly contributions. You can create and link your ORCID from the home page of the MTS by clicking on 'Modify my Springer Nature account'. For more information please visit www.springernature.com/orcid.

Yours sincerely,

Referee comments

Reviewer #2 (Cryptosporidium, host-pathogen interaction)

My remarks have been well addresses in the rebuttal letter and in the corresponding manuscript text and figures.

Regards,
Kurt Hanevik

Reviewer #3 (Cryptosporidium, drug discovery, microscopy)

Thanks for the thoughtful replies and updates as requested. I think all but one have been sufficiently addressed.

I think my comment below was misunderstood, and I would like to explain again what I meant:

"1. All fecal NIuc graphs should be line graphs rather than bar charts. The x-axis should be a number line rather than just the dpi samples were collected."

By this I meant the x-axis should be a number line, where the numbers are not a category but a number line. This means you can tell the true distance between dpi. Right now sometimes the numbers are sequential, and sometimes there are gaps. This makes it difficult to interpret because it gives the appearance that all the data points have equal lengths of time between them. But they do not, and this complicates how to interpret things, especially when mice have been dosed.

If you prefer to keep the bar graph over the number line, that seems ok, but I think the x-axis should reflect time more appropriately.

Reviewer #4 (Cryptosporidium genomics)

This study provides a compelling genetic explanation for a critical clinical challenge in the treatment of Cryptosporidium: the suboptimal efficacy of clofazimine. By integrating genetic crosses, Bulk Segregant Analysis (BSA), and rigorous gene-editing validation, the authors demonstrate that the heterogeneity of NDH2 alleles is the primary driver of resistance. This finding represents a significant paradigm shift, moving beyond the traditional view that clinical failure is solely attributable to poor drug bioavailability, and instead revealing a sophisticated evolutionary strategy of pre-existing resistance within the parasite population.

The genomics analysis section of the manuscript was specifically reviewed. The authors successfully identified a AAA deletion (Δ AA) located in a highly variable region and, using a BSA strategy, precisely linked the resistant phenotype to the NDH2 gene. This approach integrates classical genetics with high-throughput sequencing, making it highly convincing. Furthermore, a systematic reanalysis of sequencing data from global isolates further confirmed the widespread presence of NDH2 heterogeneity in natural populations. The application of genomic data and the analytical strategies employed in this study are scientifically sound and logically rigorous. The findings are well-supported by the data presented.

Version 2:

Decision Letter:

Our ref: NMICROBIOL-25103756B

18th February 2026

Dear Boris,

Thank you for submitting your revised manuscript "Widespread genomic heterogeneity at the type II NAD(P)H dehydrogenase locus predisposes *Cryptosporidium* to clofazimine resistance" (NMICROBIOL-25103756B). It has now been seen by the original referees and their comments are below. The reviewers find that the paper has improved in revision, and therefore we'll be happy in principle to publish it in Nature Microbiology, pending minor revisions to satisfy the referees' final requests and to comply with our editorial and formatting guidelines.

Thank you again for your interest in Nature Microbiology. Please do not hesitate to contact me if you have any questions.

Best wishes,

Referee comments

Reviewer #2 (*Cryptosporidium*, host-pathogen interaction)

My remarks have been well addresses in the rebuttal letter and in the corresponding manuscript text and figures.

Regards,
Kurt Hanevik

Reviewer #3 (*Cryptosporidium*, drug discovery, microscopy)

Thanks for the thoughtful replies and updates as requested. I think all but one have been sufficiently addressed.

I think my comment below was misunderstood, and I would like to explain again what I meant:

"1. All fecal NIuc graphs should be line graphs rather than bar charts. The x-axis should be a number line rather than just the dpi samples were collected."

By this I meant the x-axis should be a number line, where the numbers are not a category but a number line. This means you can tell the true distance between dpi. Right now sometimes the numbers are sequential, and sometimes there are gaps. This makes it difficult to interpret because it gives the appearance that all the data points have equal lengths of time between them. But they do not, and this complicates how to interpret things, especially when mice have been dosed.

If you prefer to keep the bar graph over the number line, that seems ok, but I think the x-axis should reflect time more appropriately.

Reviewer #4 (*Cryptosporidium* genomics)

This study provides a compelling genetic explanation for a critical clinical challenge in the treatment of *Cryptosporidium*: the suboptimal efficacy of clofazimine. By integrating genetic crosses, Bulk Segregant Analysis (BSA), and rigorous gene-editing validation, the authors demonstrate that the heterogeneity of NDH2 alleles is the primary driver of resistance. This finding represents a significant paradigm shift, moving beyond the traditional view that clinical failure is solely attributable to poor drug bioavailability, and instead revealing a sophisticated evolutionary strategy of pre-existing resistance within the parasite population.

The genomics analysis section of the manuscript was specifically reviewed. The authors successfully identified a AAA deletion (Δ AA) located in a highly variable region and, using a BSA strategy, precisely linked the resistant phenotype to the NDH2 gene. This approach integrates classical genetics with high-throughput sequencing, making it highly convincing. Furthermore, a systematic reanalysis of sequencing data from global isolates further confirmed the widespread presence of NDH2 heterogeneity in natural populations. The application of genomic data and the analytical strategies employed in this study are scientifically sound and logically rigorous. The findings are well-supported by the data presented.

Version 3:

Decision Letter:

17th March 2026

Dear Sebastian and Boris,

I am pleased to accept your Article "Genomic heterogeneity of NAD(P)H dehydrogenase predisposes *Cryptosporidium* to clofazimine resistance" for publication in Nature Microbiology. Thank you for having chosen to submit your work to us and many congratulations.

Authors may need to take specific actions to achieve compliance with funder and institutional open access mandates. If your research is supported by a funder that requires immediate open access (e.g. according to [Plan S principles](https://www.springernature.com/gp/open-science/plan-s-compliance) or the [NIH public access policy](https://www.springernature.com/gp/open-science/us-federal-agency-compliance)) then you should select the gold OA route, and we will direct you to the compliant route where possible. Because authors warrant under our subscription licensing terms that they haven't committed to licensing any version of their article under a licence inconsistent with the terms of our agreement – including the applicable embargo period – publication under the subscription model isn't suitable for authors whose funders require no embargo.

Congratulations once again and I look forward to seeing the article published.

With kind regards,

P.S. Click on the following link if you would like to recommend Nature Microbiology to your librarian
<http://www.nature.com/subscriptions/recommend.html#forms>

** Visit the Springer Nature Editorial and Publishing website at http://editorial-jobs.springernature.com?utm_source=ejP_NMicro_email&utm_medium=ejP_NMicro_email&utm_campaign=ejp_NMicro for more information about our career opportunities. If you have any questions please click [here](mailto:editorial.publishing.jobs@springernature.com).

Open Access This Peer Review File is licensed under a Creative Commons Attribution 4.0 International License, which permits use, sharing, adaptation, distribution and reproduction in any medium or format, as long as you give appropriate credit to the original author(s) and the source, provide a link to the Creative Commons license, and indicate if changes were made. In cases where reviewers are anonymous, credit should be given to 'Anonymous Referee' and the source. The images or other third party material in this Peer Review File are included in the article's Creative Commons license, unless indicated otherwise in a credit line to the material. If material is not included in the article's Creative Commons license and your intended use is not permitted by statutory regulation or exceeds the permitted use, you will need to obtain permission directly from the copyright holder.

Point-by-point response:

Please find our responses to each individual comment below, formatted in *italic blue*.

Reviewers Comments:**Reviewer #1 (Remarks to the Author):**

This is an absolutely outstanding manuscript that deserves immediate publication. The authors have made a major discovery about the mechanism of drug resistance in an important medically difficult to treat parasite called *Cryptosporidium* that is a major cause of diarrheal illness, and an opportunistic infection in HIV/AIDS patients. The work is very well and thoroughly done, and the paper is written very clearly. Despite decades of effort effective treatment for *Cryptosporidium* infection has remained elusive. Recently the anti-TB agent clofazimine entered human clinical trials for treatment of *Cryptosporidium* but failed, and the failure was attributed to poor bioavailability. However, in this manuscript the authors demonstrate that clofazimine resistance arises rapidly in *Cryptosporidium*. They use genetic crosses to identify mutations in type II NADH dehydrogenase (NDH2) that led to early truncation and loss of expression of active enzyme as the mechanism of resistance. They go on to use biochemical studies on recombinant enzyme to demonstrate that NDH2 mediates electron transfer to clofazimine suggesting prodrug activation is required for its mechanism of action. They show that unlike in other cells where NDH2 localizes to the mitochondria, in *C. parvum* it is localized to the membrane of the IMC facing the parasite cytoplasm, suggesting a unique role for this enzyme in the parasite. Finally, they demonstrate that the NDH2 locus of *C. parvum* and *C. hominis* has widespread heterogeneity, that the observed mutations are already present in the untreated populations and due to frequent sexual recombination in the parasite, they are predisposed to rapidly evolve resistance to clofazimine. I don't see anything that needs changing in what is the best manuscript and discovery I've read about in years.

We thank the reviewer for these very kind comments on our work.

Reviewer #2 (Remarks to the Author):

Review of manuscript "Widespread genomic heterogeneity at the type II NAD(P)H dehydrogenase locus predisposes *Cryptosporidium* to clofazimine resistance" by Buenconsejo et al.

Key results

In this manuscript the authors test in vitro susceptibility to clofazimine in four *Cryptosporidium* isolates and use an in vivo cross between a susceptible and resistant strain of *C. parvum* to identify the locus conferring the resistance as a two base pair deletion in the gene for type II NAD(P)H dehydrogenase. The authors further confirmed the effect of this mutation in transgenic parasites in tissue culture growth assays. They characterize the NDH2 protein function biochemically, its localization to the inner membrane complex and that the deltaAA allele does not result in complete loss of NDH2 activity. They also show NDH2 allele frequency is dynamic and that speed of resistance development is driven by initial deltaAA frequency. Examining available 68 good quality *C. parvum* and *C. hominis* whole genomes the authors show that the deltaAA allele is widespread at low frequencies in most examined isolates.

Validity

Well described reference strains were used. Authors used an elegant method to produce genetic crosses in a IFN γ -/- mouse model and selection of drug resistant strains by clofazimine treated or untreated controls. Transgenic parasites were sequenced to verify appropriate insertions. Recombinant NDH2 production enabled biochemical analyses to verify that clofazimine can be reduced by the *C. parvum* enzyme.

Significance

The manuscript identifies a usually low-frequency NDH2 allele and its capacity for clofazimine resistance, and it brings new knowledge on the role of NDH2 localization and function in *Cryptosporidium*. The authors discuss how rapid selection of *Cryptosporidium* with the deltaAA resistance allele can rapidly result in therapy failure. The authors rightly conclude that considering resistance potential is an important step in preclinical evaluation of anti-*Cryptosporidium* drugs.

Data and methodology

The authors used *C. parvum* Ila and IId derived from animal strains for their experimental studies. This is presumably done due to available methodology and reference strains, but is also pertinent in this study considering that 11 out of 22 infections failing on Clofazimine therapy in the Tam 2021 study were infected by *C. parvum*. Supplementary data in Figure 6a helps to generalize findings to *C. hominis* and human infecting *C. parvum* strains. I am only superficially familiar with the genetic bioinformatics pipeline used and refrain from commenting on it.

Suggested improvements

Line 80 – The Tam2021 study was randomized, but randomization in this small study failed and significant differences were found between the arms. It is therefore not quite correct to say “and a resulting lack of randomization”. I suggest to state instead f.ex “and significant baseline differences between study arms”.

We have rephrased this sentence in following the reviewer's suggestion

It would be interesting if authors could discuss how the long half-life of clofazimine (>30 days) will affect its drug selection capability. Most of their arguments are relevant for clofazimine effect in immunosuppressed mice or humans with chronic infections. Could they also consider the effect of the deltaAA allele on treatment of the large population of immunocompetent children normally being symptomatic for a few days only?

We use IFNg KO mice to circumvent C. parvum colonization resistance due to innate immunity – however, these mice retain T-cell function and clear the infection – they are not the mouse equivalent of HIV/AIDS patients. The initial cross occurs over a chronic time frame because we use the use of the persistent KVI strain (please see our recent paper in Cell Reports for detail on the genetics of persistence). Note that many of the subsequent mouse experiments were done using the Bunchgrass Iowa II strain which does not persist. In these experiments infection is acute, and our treatment protocol does not exceed 7 days. After these 7 days the resistant allele is already heavily enriched (see Fig 6j). Similarly, significant enrichment is measurable by bulk segregant analysis in the cross after only three days of low dose treatment (Fig. 1g and 2C). Overall selection thus appears swift and does not require persistence.

I guess what the reviewer is asking here between the lines and from a translational perspective is whether we believe clofazimine might still be helpful in immune competent children and whether resistance may not emerge there or if so then slower.

We lack expertise in pharmacokinetics and clinical patient care to speak to this question with authority and are thus a little reluctant to speculate. However, we feel that the fact that a loss of function mutation of NDH2 confers high level of resistance, and that such an allele is already widespread is cause for concern and warrants careful consideration. Even if treatment details may vary among different patient groups, the overall risk of losing clofazimine to resistance appears high.

With a starting frequency of the allele up to 5% it likely takes two or more generations to wipe out the sensitive parasites, during which one could expect temporary symptom relief and decreased shedding. The clofazimine treatment regime in the mouse model in the present manuscript transiently repressed parasite growth and shedding. In the human trial the opposite was seen in clofazimine treated patients who developed increased shedding and no change in diarrhea severity. How can this be reconciled? Could the qPCR based increase be due to shed, dead *Cryptosporidium* due to initial successful clofazimine treatment?

This is an interesting question, but one difficult to judge. The changes upon treatment in the clinical trial did not rise to statistical significance. There was a trend towards higher shedding in the treatment group, but as pointed out by the reviewer above, the arms of study ended up not being entirely random, and the treatment group experienced more severe disease even prior to treatment making it harder to interpret a small change. Indeed, the reviewer already points toward the most likely explanation for the difference, we use two different assays in two different hosts which may account for variation. Furthermore, the patients in the trial suffered from many more diseases and infections than our mice. This may further complicate comparison.

Interestingly, in the Tam 2021 study only 3 patients had *C. hominis*, while four patients had *C. meleagridis* and one had *C. viatorum*. Hopefully, post-hoc analyses of delta AA presence will be made in the actual strains obtained in that study. However, the authors show that the deltaAA is widespread, and present in most *parvum* and *hominis* genomes that they had access to, but not including *C. meleagridis*. With a recently available genome of *C. meleagridis* (Penumarthi et al. *Sci Data* 11, 1388 (2024) <https://doi.org/10.1038/s41597-024-04235-7>) it would be interesting to explore deltaAA presence also in this species.

This is a great suggestion that we implemented in the revision. We found three C. meleagridis genome data sets that fulfill the inclusion categories we used for C. parvum and hominis (high quality illumina reads with significant genome coverage). As suggested by the reviewer we aligned these genomes against the new reference and analyzed NDH2 variation. We found that those three C. meleagridis genomes lacked the delta AA indel and were 100% wildtype. We added this new data to Figure 6a and the supplemental data table.

Clarity and context

Abbreviation HA should be written out for clarity when first used.

Changed to NDH2-hemagglutinin (HA) in lines 228 as requested.

Line 100 - Does the lower potency in *C. hominis* refer to ref 11? If no, a reference should be provided here. If yes, add ref 11 here.

We added the reference to make this clearer.

Isolates are not coherently named. Line 122 - Authors refer to Buchgrass isolate. I suggest correcting to the name of the strain as given earlier here and in line 967. Related to this are the two ways the producer company is named. Is it "Bunch Grass" line 395 or "Bunchgrass" line 122. Also, they suddenly use "Sterling" to name a strain in line 113. I suggest to stick to defined names throughout the manuscript.

We thank the reviewer for noting this. We have corrected the labelling and name the strains consistently throughout the manuscript now.

In line 110-116 authors describe a clofazimine susceptibility assay without mentioning clofazimine once. It is evident from figure 1b, but would be clearer if clofazimine was mentioned f.ex in line 111.

We have added clofazimine in line 104 to enhance clarity.

Can authors clarify what is meant with "Neo" in line 124.

We have added "neomycin phosphotransferase drug-selection marker" in lines 117/118

Formatting of several references have not come out nicely and should be checked and corrected.

We fixed this.

Conclusion

In conclusion, I think this manuscript contains a wealth of new, good quality data and a sound analysis. The conclusions are well supported by the data, and makes a significant stride forward in our knowledge of *Cryptosporidium* genetic variation and metabolism. The study is a fine example of what is now possible with new laboratory tools to examine this important enteric pathogen.

We thank the reviewer for this comment.

Reviewer #3 (Remarks to the Author):

Clofazimine is a leprosy and TB drug that was evaluated for repurposing for treatment of cryptosporidiosis. It showed efficacy for cryptosporidiosis in pre-clinical work, but failed in clinical trials. The clinical trials were challenging for a variety of reasons (small number of patients, patients were immunocompromised, patients were suffering from severe disease), so it is difficult to determine exactly why this drug failed. One possibility is that the DMPK profile of the compound was insufficient for successful treatment in patients.

Here, the authors propose that the failure instead may be due to drug resistance. This work robustly identifies a mechanism of clofazimine resistance in *C. parvum*. However, there is no data from the failed clinical trial to confirm that the patients treated with clofazimine were infected with drug-resistant parasites.

First, the authors identify that different strains of *C. parvum* exhibit variation in susceptibility to clofazimine. They identify 2 strains that have a big variation in SNPs and that have different susceptibility to clofazimine. Using genetic

tools they create a hybrid strain. The hybrid strain was already produced and characterised in another publication (PMID: 40971299); I am not sure this is very clear here.

This reference is provided at the beginning of the paragraph that describes the cross and again in the context of bulk segregant analysis in the next section.

They utilise this hybrid strain to select for resistance to clofazimine. They then use BSA and QTL mapping to identify the locus that drives resistance to clofazimine. They observe that strains that have resistance to clofazimine have an indel in the 5' region of the gene, CpNDH2. They use genetic approaches to validate that this mutation gives rise to clofazimine resistance. They express recombinant NDH2 and confirm that it interacts with clofazimine. Typically, NDH2 should be located in the mitochondria, however *Cryptosporidium* do not have this organelle, and instead have a mitosome. Interestingly, NDH2 does not localise to the mitosome, but localises to the plasma membrane.

They also show that strains that carry the resistance-conferring mutation express this mutation with a lot of heterogeneity. Within the population of a resistant strain, there is both WT and mutant alleles; treatment with clofazimine selects for growth of the subpopulation that has the mutation.

The impact of this work is that the target of clofazimine for *Cryptosporidium* is identified and confirmed to be the major route to drug resistance. This is supported by innovative approaches including forward genetics, WGS, CRISPR for genetic validation, and enzymatic assays. This is very important clinically and also exciting as forward genetics has not yet been used to identify a drug target in *Cryptosporidium*. This technique could be useful to identify the target of other phenotypic hits in development for treatment, but that lack a known target.

It is really important to understand the mechanism of action of drugs, and especially when they fail. However, there is no evidence that this is the reason why the drug failed in the clinical trial. Samples from these patients were not sequenced to confirm that the *Cryptosporidium* in that clinical trial contained these resistance-conferring mutations. Therefore, I would caution that this study proves that the clinical trial failure was due to drug resistance.

We thank the reviewer for this important point. We agree that our study does not prove that the clinical trial failed due to resistance as we did not study materials from the trial. Therefore, we took care not to claim such prove. We suggested that our results "provide additional clues to the interpretation of the clinical failure of clofazimine" which we believe to be a measured and conservative statement. In the discussion we wrote that the discovery of drug selectable drug resistance already prior to implementation "casts serious doubt on the prospect of clofazimine". We removed this half sentence, and we toned down the argument at a second point.

We actually did try to measure allele frequencies in participants of the trial to obtain a direct measurement and reached out to the authors of the trial study, tragically the local lead author passed away since publication, a young physician scientist with young children. Another senior author let us know and very kindly investigated this possibility for us. Following some correspondence, we learned that in accordance with cultural sensitivity around human materials in Malawi the study protocol prescribed that all materials were to be destroyed upon completion of the initial analysis. The DNA samples obtained for diagnostic purposes might indeed have allowed us to directly measure treatment impact by amplicon sequencing; however, those samples were no longer available.

Sometimes the figures have been oversimplified and leave out key information that is required to interpret the findings.

We agree with this reviewer that clarity of figures is important, and pride ourselves on a record of high-quality illustration. There is always room for improvement, and we are more than happy to enhance clarity. We have followed many of the suggestions the reviewer makes. A few seem to be matter of preference and taste ("I'd rather see blue and gray as the main colors"). In some of those cases we stuck to the color scheme and plot type that we used in many previous publications and that we feel best represents our work.

Specific comments:

1. All fecal Nluc graphs should be line graphs rather than bar charts. The x-axis should be a number line rather than just the dpi samples were collected. This distorts the sense of time and makes it difficult to interpret the figures. This should be changed as standard for all graphs where the x-axis is time. Technical replicates should be graphed with individual data points rather than error bars. This will make the data more clear to interpret, especially for experiments where mice were treated with a drug.

We have consistently used this bar graph format to report mouse burden in numerous articles since we established this luciferase assay in 2015 including one in *Nature Microbiology* and three in *Nature*. To explore the impact of change, we replotted the data in the format the reviewer wishes us to adopt (see below) and find little difference. Overall, we prefer bar over line graph here because line graphs tend to visually interpolate values in between whereas the format we choose shows exactly what we measured and not more.

2. Report RLU/mg in all figures rather than just RLU. This has become the standard in the field.

The Materials and Methods detailed the weight of the fecal sample. We added RLU per 20mg faecal sample to the axis to clarify this further.

3. Fig 1A make the nitrogens orange in color themselves rather than insert a star.

Changed as suggested.

4. Fig.1B (and others), rather than use log microM on x-axis, use 10-x as it is much easier to understand.

Changed as suggested.

5. Fig.1B (and others), “% Luminescence of no drug” should be replaced with “% Growth inhibition (normalised to no drug control)”

This suggestion may be too bit long to fit the axis. We typically prefer growth as that is what we measure here. We changed to “% Growth of vehicle control” to enhance clarity.

6. Fig.1D all mice should be black. Use of gray mouse suggests that it’s a different kind of mouse when it’s just a sequential passage.

Changed as suggested.

7. Fig.1F Remove gray bars when CFZ treatment was '0'. This is misleading. I know the purpose is to compare to Fig.1g but actually it makes it unclear that f is essentially the "no treatment" group and g is the treatment. *We adjusted text in lines 122-124 to enhance clarity that Fig. 1f represents "no treatment" comparison group. What is shown in the figure actually reflects exactly what was done as this is not just a cage of mice that was left untreated. In the interval highlighted, mice treated with vehicle plus the indicated concentration of clofazimine, which was 0 for the first three treatments and 100 mg/kg for the last.*

8. I find the use of blue and orange in figure 1 confusing as sometimes it's meant to indicate clofazimine and sometimes it's not. I'd rather see blue and gray as the main colors, and then orange used to indicate cfz specifically (for examples the CFZ treatment in Fig.1fg). *Orange is not used to represent clofazimine it consistently represents selection.*

9. Fig.2a would benefit from having faint lines drawn at 0.5 similar to Fig.2b so it is easier to identify where regions are more similar to one of the parents. It would also be helpful to label the y-axis to indicate which parent is '0' and which is '1'. *We added a dashed line at midpoint as suggested. The reviewer is right that this graph conceptually shows whether the allele came from KVI or BG Iowa II, however, that is not exactly what the analysis measured and what was plotted here. The analysis essentially asked for each SNP whether they are identical to the BG reference or not (this is described in some detail in the Methods section). It thus has to be BG Iowa II allele frequency to be factually correct in Fig. 2a and 2b. We now explain this in the results section to enhance clarity.*

10. Fig.2d G' is lacking the prime symbol. *The prime symbol is actually not lacking by mistake here – G and G' are not the same thing and they were chosen deliberately here. G reflects statistical association of individual SNPs with the phenotype, G' reflects a collective assessment of multiple SNPs across the chromosome and is presented as a weighted moving average as indicated in the figure legend.*

11. Fig.3bd put parents on the left and progeny on the right of each panel. *It is not that we saw this difference in the parents and then analyzed the cross. We actually discovered this previously unknown allele in the cross upon selection and secondary analysis shown in these panels then demonstrated that the allele was already present (at different frequency) in the parents. To follow the logic of the text we thus show the cross first from left to right.*

12. Fig.3b is not very easy to understand what the purpose is because the sequencing maps are so small. Something zoomed in like Fig.3a for each condition would be a lot more clear. *Fig 3b is actually a zoom in of 3a for all four genomes down to the base pair level. We now say so explicitly and specify chromosome coordinates in the legend to make this clearer.*

13. Fig3. It would be helpful to label parents as KIV (CFZS) and BG (CFSR) for clarity throughout. I think the acronyms start to get complex and don't match. Sometimes it's BG and sometimes "BG Iowa II". I think it would be nice to streamline this to match. *Thanks for catching this. We now use strain names consistently in response to this and a previous comment by reviewer 2.*

14. Figure 4 is not color-blind friendly. *We have changed the red channel in figure 4h to magenta. We have also changed the red symbols in figures 4i and 4j to magenta*

15. Fig.4ad, without indication of the size you are expecting, these drawings are not so useful in context of the DNA gels. *The maps were redrawn to scale and an added scalebar allows to gauge the size of the expected fragments.*

16. Fig.4f with the bar chart it is really difficult to interpret which days post infection mice received the dosing based on the graph alone. *We changed the x-axis to "days post infection" (also changed it in in fig. 1). The grey box labeled "treat mice" indicates the days of treatment.*

17. Fig.4g y-axis of FACS data must be labelled. Gating strategy should be included as a supplement. This would also be nicely supported by microscopy.

We expanded the previous labeling of the Y axis to enhance clarity.

The gating strategy was provided in the reporting summary (this summary might not have been shared with reviewers). We now also provide this strategy in the methods section to make this easier to find. The FACS data are well supported by microscopy (see fig. 4h).

18. Fig.4h I understand that all parasites are green, but why are not all parasites red? I think this would be better explained by showing microscopy of the oocysts rather than infected culture.

*Vicia villosa lectin (VVL) labels glycoconjugates associated with the parasitophorous vacuole, extracellular stages do not stain, and early stages following invasion stain with modest intensity. This is not an absolute marker and you can observe similar patterns in many publications including in figures 1, 2, 3, and 5. of our recent publication [Wallbank, B. A., Smith, E. J., Carrasco, J. E. D., Xiao, R., Walzer, K. A., Riley, J. R., & Striepen, B. (2025). Cryptosporidium aspartyl protease 2 is required for host cell egress of merozoites and male gametes. *Molecular biology of the cell*, 36(11), ar133. <https://doi.org/10.1091/mbc.E25-06-0306>]. In contrast, mNeon is very bright and visible in all stages. We added a note to the figure legend to point this out and to provide further clarification.*

19. Figure 5 is not color-blind friendly.

We have changed all micrographs in the red channel to magenta.

20. It would be nice to quantify observations from Fig.5.

We are unsure what the reviewer specifically feels should be quantified and to which panel that should correspond as this was not specified. The figure does not make a quantitative claim. Essentially it documents that the location is to a stripe close to the surface and not to a dot for NDH2 whereas the control gene we chose to tag indeed labels a dot.

21. Fig.6a would benefit from having more details on the x-axis. As it stands I do not get a sense of how many genomic sequences were analysed (for example how many numbers of isolates, or how many isolates over time).

As described in the text of the results section and the legend each bar reflects one genome/isolate – the results text explicitly specifies the number, the x-axis also allows to gauge that number. Extensive meta data including accession number, time of collection, location, and time are provided in detail in a data table supplied in the extended material. We now explicitly refer to these meta data in the results section to make this easier to find.

22. Fig.6g Why not use quantitative proteomics instead?

Western blotting is a well-established approach to document presence or absence of proteins. We have used quantitative proteomic experimentation in the past (see e.g. 2023 Cell Host Microbe article by Guerin et al.). This may not be as straightforward and superior as the reviewer believes. Infected cultures are dominated heavily by human proteins – this is a small parasite in an ocean of large mammalian cells. To measure a rare protein (as in one that is only marginally expressed due to an indel) is truly challenging. We feel the question is not so much why we may not use other approaches but whether the approach we have chosen supports our claim that the mutant protein is present at a much lower level. We believe our Western blot does that and we used microscopy as a second orthogonal measurement. We find that both agree.

23. Fig.6lm and Fig.3c why don't these have data points plotted as in Fig.6k?

Because these are in vivo data that reflect measurement across and entire cage of mice from which pooled feces were collected. Panel K show in vitro data where each dot reflects a different well.

A few comments on the experimentation:

It would be helpful to perform some structural work to understand how NHD2 and clofazimine interact. Can the authors perform some docking studies? Or mutate regions of the recombinant protein and ablate drug interactions?

The focus of this study was not the biochemical detail of drug interaction but the overall mechanism and the genetic and genomic underpinnings of susceptibility. These are interesting ideas diving into the mechanism of clofazimine action that the biochemist in the team around Dr. Zhu may pursue in future studies.

Fig.6h and i: the error bars are very big on some of these samples and the line is not accurate in representing the data. I do not think you can calculate an accurate IC50 from this data.

We agree that there is variance. Part of that stems from the very strong change in drug susceptibility due to the mutation which forced us to compare this over multiple orders of magnitude. To be honest, we worked hard on these

measurements which requires the comparison of multiple strains, and this is as tight as we could get it. We are happy to acknowledge variance (which is on display in the figure as we show all data) and therefor now say so explicitly in the legend. However, the claim that mutation dramatically shifts the IC50 is well supported by the data shown.

A few comments on the writing:

I am not sure that NDH2 “is confined to one side of the IMC” (line 359). This requires further evidence. I am not sure this can be determined using the methods applied here.

We are not really voicing a claim here – we are simply describing the observation – green label is consistently found on the inner side of the IMC membrane. But as we had to cut words anyway, we removed this sentence.

I find the phrase NDH2 “act as a functional diploid” (line 389) problematic. This is because Plasmodium and other parasites use copy-number variation as mechanism for drug resistance. The use of the word diploid here suggests a similar mechanism could occur in Cryptosporidium, where clearly there is just variation within the population, not a single parasite having a duplicated loci or a duplicated chromosome. I think this phrase should be removed for clarity.

We rephrased this statement to enhance clarity.

Point-by-point response:

Please find our responses to each individual comment below, formatted in *italic blue*.

Referee comments

Reviewer #2 (Cryptosporidium, host-pathogen interaction)

My remarks have been well addresses in the rebuttal letter and in the corresponding manuscript text and figures.

Regards,
Kurt Hanevik

We acknowledge the reviewer's comment.

Reviewer #3 (Cryptosporidium, drug discovery, microscopy)

Thanks for the thoughtful replies and updates as requested. I think all but one have been sufficiently addressed.

I think my comment below was misunderstood, and I would like to explain again what I meant:

"1. All fecal Nluc graphs should be line graphs rather than bar charts. The x-axis should be a number line rather than just the dpi samples were collected."

By this I meant the x-axis should be a number line, where the numbers are not a category but a number line. This means you can tell the true distance between dpi. Right now sometimes the numbers are sequential, and sometimes there are gaps. This makes it difficult to interpret because it gives the appearance that all the data points have equal lengths of time between them. But they do not, and this complicates how to interpret things, especially when mice have been dosed.

If you prefer to keep the bar graph over the number line, that seems ok, but I think the x-axis should reflect time more appropriately.

We really don't make any kinetic argument here, we just report the amplitude of luciferase activity. We do not show every day as we did not measure every day. To accommodate the reviewer's point we now state in the legend that there were no measurements on day x, y and z.

Reviewer #4 (Cryptosporidium genomics)

This study provides a compelling genetic explanation for a critical clinical challenge in the treatment of Cryptosporidium: the suboptimal efficacy of clofazimine. By integrating genetic crosses, Bulk Segregant Analysis (BSA), and rigorous gene-editing validation,

the authors demonstrate that the heterogeneity of NDH2 alleles is the primary driver of resistance. This finding represents a significant paradigm shift, moving beyond the traditional view that clinical failure is solely attributable to poor drug bioavailability, and instead revealing a sophisticated evolutionary strategy of pre-existing resistance within the parasite population.

The genomics analysis section of the manuscript was specifically reviewed. The authors successfully identified a AAA deletion (Δ AAA) located in a highly variable region and, using a BSA strategy, precisely linked the resistant phenotype to the NDH2 gene. This approach integrates classical genetics with high-throughput sequencing, making it highly convincing. Furthermore, a systematic reanalysis of sequencing data from global isolates further confirmed the widespread presence of NDH2 heterogeneity in natural populations. The application of genomic data and the analytical strategies employed in this study are scientifically sound and logically rigorous. The findings are well-supported by the data presented.

We thank the reviewer for these kind comments.